# A CK2 and SUMO-dependent, PML NB-involved regulatory mechanism controlling BLM ubiquitination and G-quadruplex resolution

Shichang Liu [1], Erin Atkinson[1,2], Adriana Paulucci-Holthauzen[1] & Bin Wang [1,2] ✉

The Boom syndrome helicase (BLM) unwinds a variety of DNA structures such as Guanine (G)-quadruplex. Here we reveal a role of RNF111/Arkadia and its paralog ARKL1, as well as Promyelocytic Leukemia Nuclear Bodies (PML NBs), in the regulation of ubiquitination and control of BLM protein levels. RNF111 exhibits a non-canonical SUMO targeted E3 ligase (STUBL) activity targeting BLM ubiquitination in PML NBs. ARKL1 promotes RNF111 localization to PML NBs through SUMO-interacting motif (SIM) interaction with SUMOylated RNF111, which is regulated by casein kinase 2 (CK2) phosphorylation of ARKL1 at a serine residue near the ARKL1 SIM domain. Upregulated BLM in ARKL1 or RNF111-deficient cells leads to a decrease of G-quadruplex levels in the nucleus. These results demonstrate that a CK2- and RNF111-ARKL1-dependent regulation of BLM in PML NBs plays a critical role in controlling BLM protein levels for the regulation of G-quadruplex.

The Bloom syndrome helicase (BLM) has multiple roles in many aspects of DNA metabolism, unwinding duplex DNA and a variety of complex DNA structures in DNA replication, recombination, transcription, and telomere maintenance[1–3]. Among the many types of DNA structures, BLM has been shown particularly active in unwinding guanine (G)-quadruplex (G4) DNA in vitro[4,5]. G4 is formed in G-rich DNA region containing four stretches of three or more consecutive Gs that fold into a secondary structure[6–8]. Sequences that form G4 are widely present in the genome and G4 formation is linked with key biological processes ranging from transcription, translation, telomere homeostasis, genomic instability and cancer[6–8]. It has been indicated that BLM binds and unwinds G4 playing an important role in suppressing G4 levels in the cell[3,9,10].

BLM primarily resides in Promyelocytic Leukemia Nuclear Bodies (PML NBs) in unperturbed cells[11–13]. PML NBs are membrane-less organelles forming a sphere of 0.1-1 um in diameter and are present in most mammalian cell nuclei[14,15]. BLM's localization to PML NBs is dependent on both the SUMOylation of BLM and the two SUMO interacting motifs (SIMs)[16,17]. PML NBs have been implicated in a variety of biological processes such as senescence, apoptosis, and genome maintenance[14,15]. SUMOylation and SUMO-SIM interaction play essential roles during the formation and function of PML NBs[13,18]. The biological significance of PML NBs in the regulation of BLM function is still not clear.

SUMO-targeted E3 ubiquitin ligases (STUBLs) are a group of SIM-containing RING-type ubiquitin ligases that selectively recognize and ubiquitinate SUMO-modified proteins[19–23]. RNF111 (also known as Arkadia) has been indicated as a STUBL since it possesses three successive SIMs at its N-terminal half of the protein and a RING domain at its C-terminus, capable of binding to SUMO2/3 chains for their ubiquitination in a SIM- and RING-dependent manner[24–28]. It has been shown that the SIMs of RNF111 are required for RNF111 localizing to PML NBs and binding to SUMOylated PML[26]. In addition, the E3 ligase activity of RNF111 is required for the ubiquitination and degradation of

[1]Department of Genetics, The University of Texas M.D. Anderson Cancer Center, Houston, TX 77030, USA. [2]Genetics and Epigenetics Program, The MD Anderson Cancer Center and UT Health Graduate School of Biomedical Sciences, Houston, TX 77030, USA. ✉e-mail: bwang3@mdanderson.org

several regulators in the TGF-β signaling pathway[29–33]. The function of RNF111 and the regulation of RNF111-mediated degradation of SUMO-modified substrates remains to be further understood.

RNF111 has an N-terminal paralog Arkadia-like 1 (ARKL1, also known as C18orf25), which shows similarity to the amino-terminal half of RNF111 and contains two SIMs that correspond to the SIM2 and SIM3 of RNF111[28]. ARKL1 has been identified as a SUMO2 binding protein by GST-SUMO2 affinity capture and a yeast two-hybrid screen against SUMO[34,35]. It also has been shown that ARKL1 can localize to PML NBs, binds to casein kinase 2 (CK2) regulatory β subunit, and is involved in Epstein-Barr virus reactivation[36,37]. The function of ARKL1 remains largely uncharacterized.

In this study, we identify a CK2- and RNF111-ARKL1-dependent and PML-NB-involved regulatory mechanism controlling BLM ubiquitination and G4 resolution. RNF111 exhibits a noncanonical STUBL activity that requires SUMO-SIM-dependent interacion with ARKL1 as well as PML NBs localization for the ubiquitination and degradation of BLM. ARKL1 SIM domains bind to SUMOylated RNF111 at lysine 237 and 238 residues, promoting RNF111 localization to PML NBs. In addition, CK2-mediated phosphorylation of ARKL1 on a serine residue nearby the first SIM of ARKL1 promotes the SIM-SUMO interaction of ARKL1 and RNF111, facilitating RNF111 localization to PML NBs and ubiquitination of BLM. The PML NBs localization of RNF111 and BLM are critical for RNF111-dependent ubiquitination and protein degradation of BLM. Loss of RNF111 or ARKL1 or inhibition of CK2 leads to reduced G4 levels in the nucleus due to uncontrolled BLM protein levels. We proposes a model that BLM level in the cell is limited by RNF111-ARKL1-mediated ubiquitination and degradation of BLM in PML NBs under the regulation of CK2 activity for maintaining G4 levels.

## Results

### RNF111 ubiquitinates BLM and limits BLM protein level

We found that BLM protein level is upregulated in various cells depleted of RNF111 by siRNA or *RNF111* knockout (KO) (Fig. 1a). The elevated BLM protein level in *RNF111* KO cells was reduced by complementation with RNF111 but not empty vector (Fig. 1b). Using cyclohexmide (CHX) chasing assay monitoring protein stability of BLM in the control and siRNF111 treated cells, we found that BLM protein stability is increased in RNF111 depleted cells (Fig. 1c). These data indicate that RNF111 plays a role in limiting BLM protein levels in cells through regulating BLM protein stability. We then investigated whether RNF111 E3 ligase activity is involved in regulating BLM ubiquitination and degradation. We treated cells with MG132, a proteasome inhibitor, and carried out endogenous BLM immunoprecipitation (IP) under denature condition. We found that BLM ubiquitination was reduced when cells were depleted of RNF111 (Fig. 1d). Furthermore, wild-type RNF111 (RNF111$^{WT}$), but not a catalytic inactive mutant of RNF111 with two of the cysteine residues in the RING domain mutated to serine residues (RNF111$^{CS}$), restored BLM ubiquitination and reduced elevated BLM levels in *RNF111* KO cells (Fig. 1e and Supplementary Fig. 1a). In an in vitro ubiquitination assay, purified BLM was ubiquitinated by RNF111$^{WT}$ but not RNF111$^{CS}$ (Supplementary Fig. 1b). Thus, the ubiquitination of BLM depends on the catalytic activity of RNF111. In addition to the regulation of BLM ubiquitination, we found that RNF111 appears to self-ubiquitinate and regulate its own stability. Immuno-precipitated Flag-tagged RNF111$^{WT}$ under denature condition was detected with polyubiquitin conjugates and inactivation of the catalytic activity (RNF111$^{CS}$) abolished the ubiquitination (Fig. 1f). Together, these data indicate that RNF111 is a E3 ligase that ubiquitinates BLM as well as self-ubiquitinates.

RNF111 has been indicated as a STUBL that binds to SUMOylated substrate through its SIMs for ubiquitination of the substrate[24–28]. We investigated whether RNF111 interacts with BLM through its SIMs. By immunoprecipitating HA-tagged RNF111 and comparing WT and SIMs mutant of RNF111 (RNF111$^{SIM*}$), we found that HA-tagged RNF111$^{SIM*}$

mutant exhibited similar degree of binding to BLM as RNF111$^{WT}$ did, indicating that the SIM domains of RNF111 are not required for RNF111 association with BLM (Fig. 1g and Supplementary Fig. 1c). We also tested whether RNF111-BLM interaction depends on the SUMOylation of BLM. Previously, it has been shown that BLM SUMOylation sites mutant (K317/331 R) or SIMs mutant (SIM*) fails to be SUMOylated[16,17]. We found that GFP-tagged BLM WT, SIM*, or K317R/K331R mutant showed similar binding to endogenous RNF111, indicating that BLM SUMOylation is not required for BLM binding to RNF111 (Fig. 1h). Thus, although RNF111 ubiquitinates BLM and regulates BLM protein stability, it does not appear to be a canonical STUBL since its binding to the substrate BLM does not directly involve a SIM-SUMO interaction.

### ARKL1 promotes BLM and RNF111 ubiquitination and degradation

ARKL1 is homologous to the N terminus of RNF111, containing two SIM domains (Fig. 2a)[24]. We found that depletion of ARKL1 by siRNAs or knockout also led to increased BLM protein levels in cells (Fig. 2b). This is correlated with enhanced BLM protein stability in *ARKL1* knockdown cells shown by the cycloheximide chasing experiment (Fig. 2c). In addition, BLM ubiquitination detected by western blot of immuno-precipitated BLM under denature condition was much reduced in the absence of ARKL1 (Fig. 2d). Thus, ARKL1 plays a role in promoting BLM ubiquitination and protein degradation. ARKL1 also appears to regulate RNF111 self-ubiquitination and protein stability since loss of ARKL1 led to increased RNF111 protein stability (Fig. 2e), reduced RNF111 ubiquitination and elevated RNF111 protein levels (Fig. 2f and Supplementary Fig. 2a). Moreover, BLM and RNF111 were found in the immunoprecipitates of ARKL1, indicating that ARKL1 interacts with BLM and RNF111 (Fig. 2g and Supplementary Fig. 2b). Thus, ARKL1 forms a complex with RNF111 and BLM, promoting RNF111 self-ubiquitination and RNF111-catalyzed ubiquitination of BLM and protein degradation.

### RNF111 is SUMOylated and interacts with the SIM domains of ARKL1

We found that RNF111 mediates the interaction of ARKL1 and BLM since knockdown of RNF111 largely abolished the interaction of ARKL1 and BLM (Fig. 3a). This is confirmed by the proximity ligation assay (PLA) showing that the interaction of ARKL1 and BLM indicated by PLA foci disappeared in *RNF111* KO cells (Fig. 3b). The interaction of ARKL1 with RNF111, however, is independent of BLM since knockdown of BLM had little effect on the interaction of Flag-tagged ARKL1 with RNF111 (Supplementary Fig. 3a).

When we examined the interaction of ARKL1 and RNF111, we found that the SIMs of RNF111 are not required for RNF111 binding to ARKL1 since RNF111$^{SIM*}$ was still able to interact with ARKL1 (Supplementary Fig. 3b). Instead, the SIM mutant of ARKL1 (ARKL1$^{SIM*}$) exhibited reduced interaction with RNF111 or BLM, indicating that the SIM domains of ARKL1 are critical for the ARKL1 interaction with RNF111 and BLM (Fig. 3c and Supplementary Fig. 3c). This is confirmed by the PLA assay showing that HA-ARKL1 and RNF111 interaction PLA foci are detected in *ARKL1* KO cell expressing wild-type ARKL1 but nearly undetectable in cells expressing ARKL1$^{SIM*}$ (Fig. 3d). Together, these results indicate that RNF111-ARKL1 interaction is mediated by the SIMs of ARKL1.

We then hypothesized that RNF111 is SUMOylated and that the SUMOylated RNF111 interacts with the SIM domains of ARKL1. To determine the SUMO and SIM interaction, we seek to identify the SUMOylation sites on RNF111. SUMOylation is frequently targeted to lysine residue (K) in the K-X-E/D motifs or an inverted version of this sequence on a substrate[38]. We noticed that all the RNF111 lysine residues with or without the K-X-E/D motif are present at the N-terminus and C-terminus of the protein. We therefore generated a series of deletion mutants that span all the lysine residues in RNF111 sequence

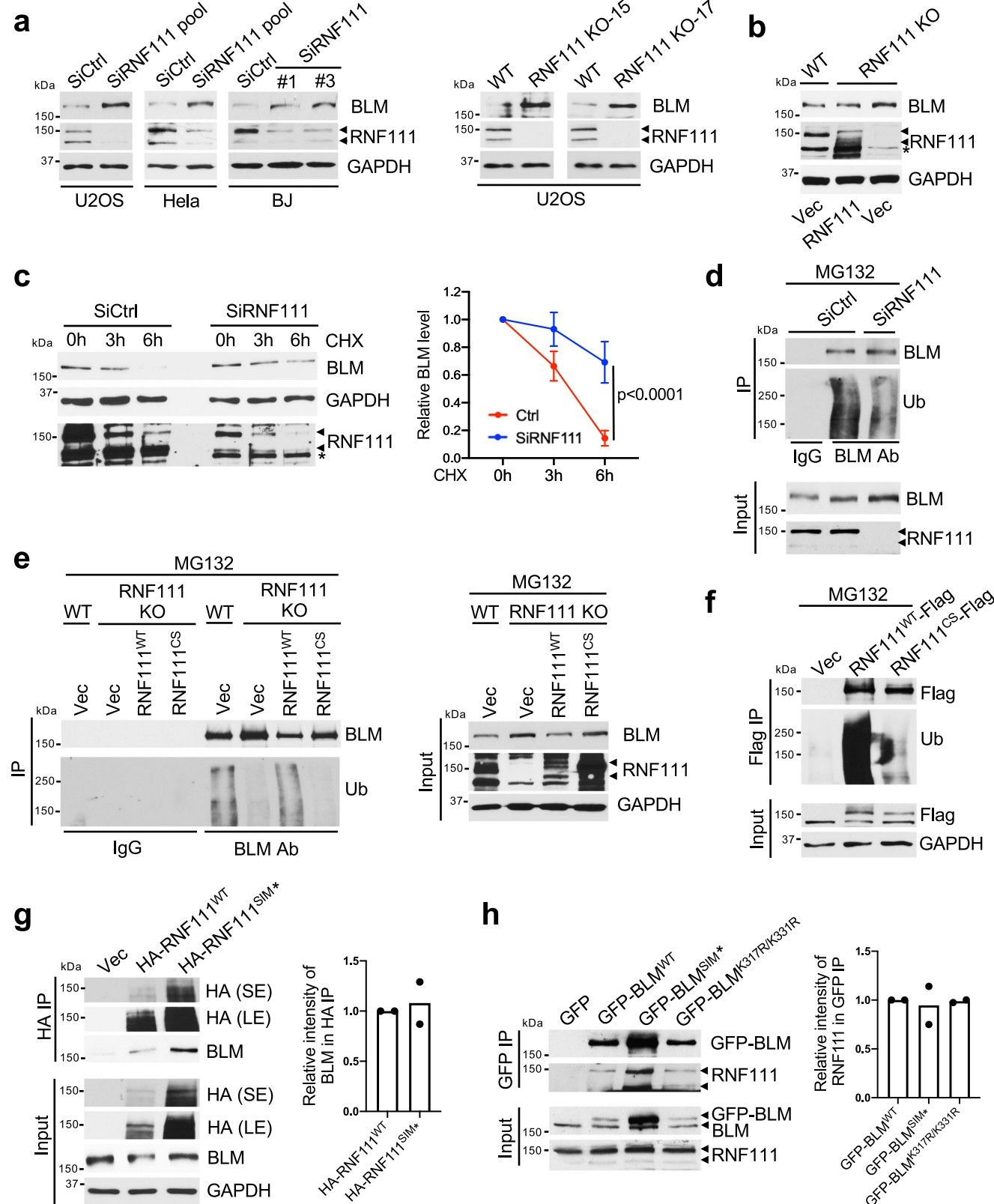

(Fig. 3e). We reasoned that if SUMOylation of RNF111 mediates the interaction of RNF111 with ARKL1 SIMs, mutation of the SUMOylation sites should decrease the binding of RNF111 and ARKL1. We thus tested the interaction of the deletion mutants of RNF111 with ARKL1. We found that deletion of 214–246 amino acids greatly reduced the interaction of RNF111 with ARKL1 (Supplementary Fig. 3d–g). Furthermore, whereas immunoprecipitated HA-tagged RNF111$^{WT}$ under

denature condition was detected with polySUMOylation with SUMO2/3 conjugates, Δ214–246 mutant showed greatly reduced SUMOylation, indicating that majority of the SUMOylation on RNF111 occurs at sites within the deleted region (Fig. 3f). We further identified that lysine 237 (K237) and 238 (K238) residues within the region are likely to be the SUMOylation sites. K238 is a SUMOylation consensus site and well conserved in various species (Fig. 3g). Mutation of both residues

**Fig. 1 | RNF111 ubiquitinates BLM and regulates BLM protein level. a** Loss of RNF111 leads to increased BLM protein levels. RNF111 bands are indicated by arrows. Cells transfected with control (Ctrl) or RNF111 siRNAs or two independent clones of *RNF111* knockout (KO) cells (KO-15 and KO-17) are shown. **b** Expression of RNF111 but not empty vector (Vec) reduces elevated BLM levels in RNF111 KO U2OS cells. "*" non-specific band. **c** Increased BLM protein stability in RNF111 knockdown cells. U2OS cells transfected with indicated siRNAs were treated with cycloheximide (CHX) (100 μg/ml). "*" non-specific band. The relative BLM level at each time point normalized by GAPDH and relative to that at 0 h is quantified and shown with mean ± SEM (*n* = 3 independent experiments). Two way Anova was used for statistics. Band intensity was measured using Image J. **d** RNF111 knockdown leads to decreased BLM ubiquitination. U2OS cells transfected with siRNAs were treated with MG132 (10 μM, 4 h). Immunoprecipitation (IP) was performed under denature

condition. **e** BLM ubiquitination is dependent on RNF111 catalytic activity. The IPs were performed under denature condition using lysates from WT or *RNF111* KO U2OS cells expressing indicated constructs and treated with MG132. **f** RNF111 catalytic activity is required for its self-ubiquitination. The IPs were performed under denature condition using lysates from 293 T cells expressing indicated constructs and treated with MG132. **g** The SIM mutant of RNF111 interacts with BLM. 293T cells expressing indicated constructs were used. The relative BLM level in the HA immunoprecipitates (normalized by immunoprecipitated HA-RNF111) is quantified (*n* = 2 independent experiments). **h** The SIMs and SUMOylation mutant of BLM interacts with RNF111. 293 T cells expressing indicated constructs were used. The relative RNF111 level present in the GFP immunoprecipitates (normalized by immunoprecipitated GFP-BLM) is quantified (*n* = 2 independent experiments). Source data are provided as a Source Data file.

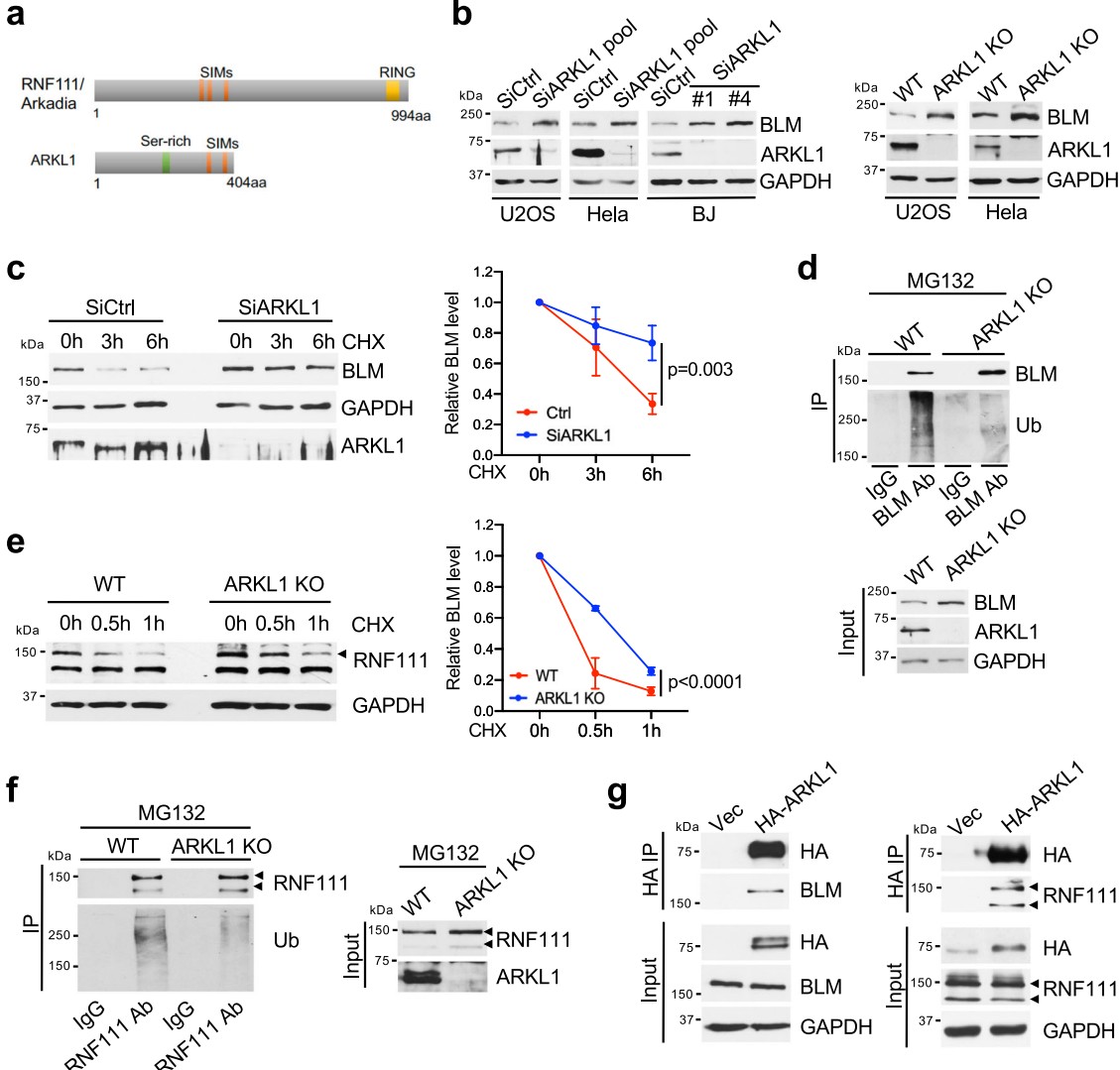

**Fig. 2 | ARKL1 promotes BLM ubiquitination and regulates BLM protein level. a** A schematic of RNF111 and its N-terminal paralog ARKL1. **b** Loss of ARKL1 leads to an increase of BLM protein level in various cell lines. **c** Increased BLM protein stability in ARKL1 knockdown cells. U2OS cells transfected with indicated siRNAs were treated with cycloheximide (CHX) (100 μg/ml). The relative BLM level at each time point (normalized by GAPDH and relative to that at 0 h) is quantified and shown with mean ± SEM (*n* = 3 independent experiments). Two way Anova was used for statistics. Band intensity was measured using Image J. **d** Loss of ARKL1 leads to decreased BLM ubiquitination. The IPs were performed under denature condition using lysates from WT or *ARKL1* KO Hela cells treated with MG132 (10 μM, 4 h).

**e** Loss of ARKL1 leads to increased RNF111 protein stability. The relative RNF111 level (indicated by arrow) at each time point after CHX treatment (normalized by GAPDH and relative to the beginning of treatment) is quantified and shown with mean ± SEM (*n* = 3 independent experiments). Two way Anova was used for statistics. **f** Loss of ARKL1 leads to reduced RNF111 self-ubiquitination. The IPs were performed under denature condition using lysates from WT or *ARKL1* KO Hela cells treated with MG132. **g** ARKL1 interacts with both BLM and RNF111. HA IPs were performed using lysates from U2OS cells expressing vector or HA-ARKL1. Source data are provided as a Source Data file.

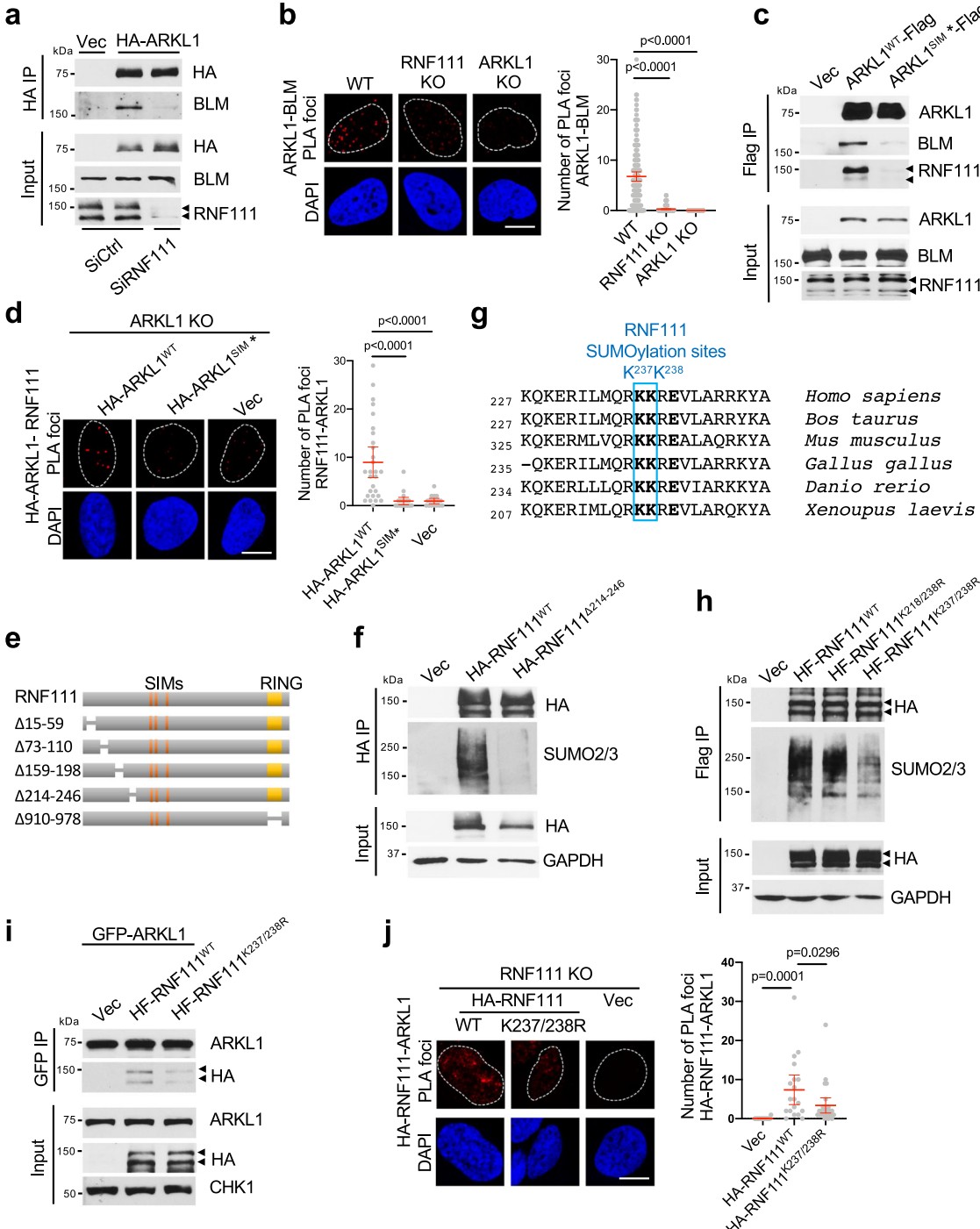

**Fig. 3 | ARKL1 interacts with RNF111 in a SIM-SUMO dependent manner.**
**a** RNF111 knockdown disrupts the interaction of ARKL1 and BLM. HA IP was performed with lysates of U2OS cells expressing vector, HA-ARKL1 and treated with indicated siRNAs. **b** Loss of RNF111 abolishes the interaction of ARKL1 and BLM in proximity ligation assay (PLA). Quantifications of WT ($n = 170$), *RNF111* KO ($n = 189$), *ARKL1* KO ($n = 115$) U2OS cells from two biological replicates are shown. **c** The SIMs of ARKL1 are required for ARKL1 interaction with both RNF111 and BLM. Lysates from 293 T cells expressing indicated constructs were used. **d** The co-localization/interaction of RNF111 and ARKL1 is dependent on the SIMs of ARKL1. PLA using antibodies against HA and RNF111 are qunatified for *ARKL1* KO U2OS cells complemented with HA-ARKL1$^{WT}$ ($n = 28$), HA-ARKL1$^{SIM*}$ ($n = 20$) or vector ($n = 17$) from two biological replicates. **e** A schematic of RNF111 deletion mutants generated. **f** RNF111$^{\Delta214-246}$ deletion mutant exhibits decreased SUMOylation. HA IPs were

performed under denature condition using lysates of 293 T cells expressing indicated constructs. **g** Identification of RNF111 SUMOylation sites at K237 and K238. Sequences flanking the SUMOylation sites from various species are aligned. **h** RNF111$^{K237/238R}$ mutant exhibits reduced SUMOylation. Flag IP was performed under denature condition using lysates of 293T cells expressing indicated constructs. **i** The interaction of RNF111 and ARKL1 requires RNF111 K237/238. 293T cells co-expressing GFP-ARKL1 and HA-Flag-tagged RNF111$^{WT}$ or RNF111$^{K237/238R}$ mutant were used. **j** The co-localization/interaction of RNF111 and ARKL1 requires RNF111 K237/238. PLA using antibodies to HA and ARKL1 are quantified for *RNF111* KO U2OS cells complemented with vector ($n = 19$), HA-RNF111$^{WT}$ ($n = 18$), and HA-RNF111$^{K237/238R}$ ($n = 28$). For **b**, **d**, **j** quantification is shown with mean ± 95% confidence interval (CI), One-way ANOVA with Sidak's correction was used for statistics, scale bar, 10 μm. Source data are provided as a Source Data file.

(K237/238 R) but not each individual residue (K237R or K238R) reduced RNF111 SUMOylation (Fig. 3h and Supplementary Fig. 3h, i). In addition, although K237/238 R mutant retains the ability to interact with BLM (Supplementary Fig. 3j), its interaction with ARKL1 is reduced shown by the co-immunoprecipitation and PLA (Fig. 3i, j).

Together, these results indicate that RNF111 is SUMOylated at K237/238 residues and that RNF111 interacts with ARKL1 in a SUMO-SIM-dependent manner.

## ARKL1 promotes RNF111 localization to PML NBs

Previously, it has been shown that RNF111 and ARKL1 can localize to PML NBs[26,37]. In U2OS cells expressing ARKL1-GFP and mcherry-PML, ARKL1-GFP is mainly in the nucleoplasm and forms a few foci that overlap with PML NBs marked by mcherry-PML (Fig. 4a), consistent with the previous finding that ARKL1 can localize to PML NBs[37].

We then investigated the recruitment of ARKL1 to PML NBs using fluorescence recovery after photobleaching (FRAP). After photobleaching, the fluorescence of ARKL1-GFP in PML NBs immediately recovered with a half-time fluorescence recovery ($T_{1/2}$) within 2 s (Fig. 4a). This demonstrates a rapid and dynamic exchange of ARKL1 within PML NBs and with the surrounding nucleoplasm for the recruitment of ARKL1 to PML NBs.

We also investigated the accumulation of RNF111 to PML NBs using live cell imaging and FRAP. Since wild-type RNF111 self-ubiquitinates and is unstable (Fig. 1f)[26], GFP-tagged RNF111[WT] expression is too low for a detection using live-cell imaging. We therefore used GFP-RNF111[CS] mutant which is unable to self-ubiquitinate and is more stable. GFP-RNF111[CS] interacts with ARKL1 and localizes to PML-NBs as WT does (Fig. 4b and Supplementary Fig. 4a)[26]. Upon bleach, GFP-RNF111[CS] exhibits a dynamic nature in its localization to PML NBs with a half-time recovery of fluorescence intensity much slower than that of ARKL1 (Fig. 4b). In addition, RNF111 appears less mobile than ARKL1 showing a smaller mobile fraction during the fluorescence recovery in PML NBs when compared to that of ARKL1 (Fig. 4c).

Since both RNF111 and ARKL1 can localize to PML NBs, it intrigued us to investigate whether the interaction of RNF111 and ARKL1 can occur at PML NBs. We noticed that a number of HA-RNF111 foci colocalize with ARKL1-GFP foci and PML NBs when both proteins are co-expressed in *RNF111* KO cells (Supplementary Fig. 4b). Examination of the interaction of RNF111 and ARKL1 using PLA in cells expressing GFP-PML showed that a significant portion of RNF111-ARKL1 PLA foci (~30%) colocalize with PML NBs, indicating that the interaction of RNF111 and ARKL1 can occur at PML NBs (Fig. 4d).

We found that loss of ARKL1 led to a more distributive pattern of RNF111 in the nucleus with much decreased number of nuclear foci that colocalize with PML NBs (Fig. 4e). The decreased localization of RNF111 to PML NBs is further confirmed by PLA showing that RNF111-PML PLA foci are significantly reduced in *ARKL1* KO cells (Fig. 4f). Reduced colocalization of RNF111 with PML NBs detected by PLA was also seen in ARKL1-deficient BJ cells (Supplementary Fig. 4c). Thus, ARKL1 plays a critical role in promoting RNF111 localization to PML NBs. The ARKL1 localization to PML NBs, however, is independent of RNF111 since HA-ARKL1 still forms foci that colocalize with PML in the absence of RNF111 (Supplementary Fig. 4d). We found that the localization of ARKL1 to PML NBs is dependent on the SIMs since ARKL1[SIM*] mutant fails to localize to PML NBs (Supplementary Fig. 4e). This is likely due to the reason that the SIM domains of ARKL1 are also required to bind to PML (Supplementary Fig. 4f). Together, the interaction of RNF111 and ARKL1 can occur in PML NBs and ARKL1 promotes RNF111 localization to PML NBs.

We then investigated whether SUMO-SIM mediated RNF111 and ARKL1 interaction is required for the localization of RNF111 to PML NBs. The PLA of RNF111-PML in *ARKL1* KO cells complemented with ARKL1[WT] or ARKL1[SIM*] showed that ARKL1[SIM*] mutant exhibited reduced ability to

promote RNF111 localization to PML NBs (Fig. 4g and Supplementary Fig. 4g). The SUMOylation of RNF111 also appears to be critical for the localization of RNF111 to PML NBs. HA-tagged SUMOylation-deficient RNF111[K237/238R] mutant, when expressed in RNF111 KO cells, showed a more distributive pattern with reduced number of foci that colocalize with PML NBs (Fig. 4h). Therefore, ARKL1 promotes RNF111 localization to PML NBs through its SIM domains interacting with SUMOylated RNF111.

## RNF111-mediated ubiquitination of BLM is associated with PML NBs

In unperturbed cells, a significant portion of BLM resides in PML NBs, forming foci that colocalize with PML NBs[11,12]. In addition to increased overall nuclear intensity of BLM and BLM foci number in *RNF111* KO and *ARKL1* KO cells, we observed that BLM intensity in PML NBs was much enhanced (Fig. 5a and Supplementary Fig. 5a, b). In addition, depletion of PML also leads to increased BLM protein levels (Supplementary Fig. 5c). These led us to hypothesize that RNF111-mediated ubiquitination and protein degradation of BLM may occur in PML NBs. We first examined whether RNF111 and BLM interaction occurs in PML NBs. By carrying out PLA of RNF111 and BLM in cells expressing GFP-PML, we noticed that a significant percentage (~30%) of the RNF111-BLM PLA foci colocalize with GFP-PML NB, indicating that RNF111 and BLM interaction can occur in PML NBs (Fig. 5b).

To determine whether PML NBs localization plays a critical role in BLM ubiquitination, we asked the question what happens to BLM ubiquitination when RNF111 or BLM does not localize to PML NBs. First, we examined RNF111 SIM* mutant which has been shown deficient in localizing to PML NBs (Supplementary Fig. 5d)[26]. By comparing *RNF111* KO cells complemented with RNF111[WT] or RNF111[SIM*], we found that BLM ubiquitination was profoundly reduced in RNF111[SIM*]-complemented cells (Fig. 5c). In addition, like catalytic inactive mutant RNF111[CS], RNF111[SIM*] was not able to reduce the elevated BLM levels in *RNF111* KO as RNF111[WT] did (Fig. 5d). We also noticed that RNF111[SIM*] failed to ubiquitinate itself as RNF111[WT] did, suggesting that PML NB localization is also required for RNF111 self-ubiquitination (Fig. 5e). Together, these data suggest that RNF111 localization to PML NBs is critical for RNF111's catalytic activity toward BLM ubiquitination and self-ubiquitination.

We then examined whether BLM mutant that does not localize to PML NBs can be ubiquitinated. The SIM domains of BLM are required for BLM's localization to PML NBs and BLM SIM* mutant fails to localize to PML NBs[17]. The SUMOylation-deficient BLM K317R/K331R mutant is also defective in the localization to PML NBs (Supplementary Fig. 5e)[16]. We found that, although both BLM[SIM*] and BLM[K317/K331R] mutants retain the ability to interact with RNF111 in vitro in the co-immunoprecipitation experiments (Fig. 1h), they failed to be ubiquitinated (Fig. 5f). This indicates that the localization of BLM to PML NBs plays a critical role in the ubiquitination of BLM.

Since ARKL1 interacts with RNF111 and promotes RNF111 localization to PML NBs, we assessed the effect of ARKL1 and RNF111 interaction on BLM ubiquitination. We found that ARKL[SIM*] mutant, which is unable to interact with RNF111, failed to restore BLM ubiquitination or reduce the elevated BLM levels in *ARKL1* KO cells as ARKL1[WT] did (Fig. 5g, h). The SUMOylation mutant of RNF111, RNF111[K237/238R], which is defective in binding to ARKL1, also was unable to restore BLM ubiquitination in RNF111 KO cells (Fig. 5i). Thus, the SIM-SUMO-mediated ARKL1 and RNF111 interaction is required for BLM ubiquitination. It is consistent with the idea that ARKL1-dependent RNF111 localization to PML NBs is critical for BLM ubiquitination.

Together, these data indicate that RNF111-mediated BLM ubiquitination and degradation probably occurs in PML NBs and ARKL1 facilitates it through SIM-SUMO dependent interaction with RNF111 promoting RNF111 localization to PML NBs.

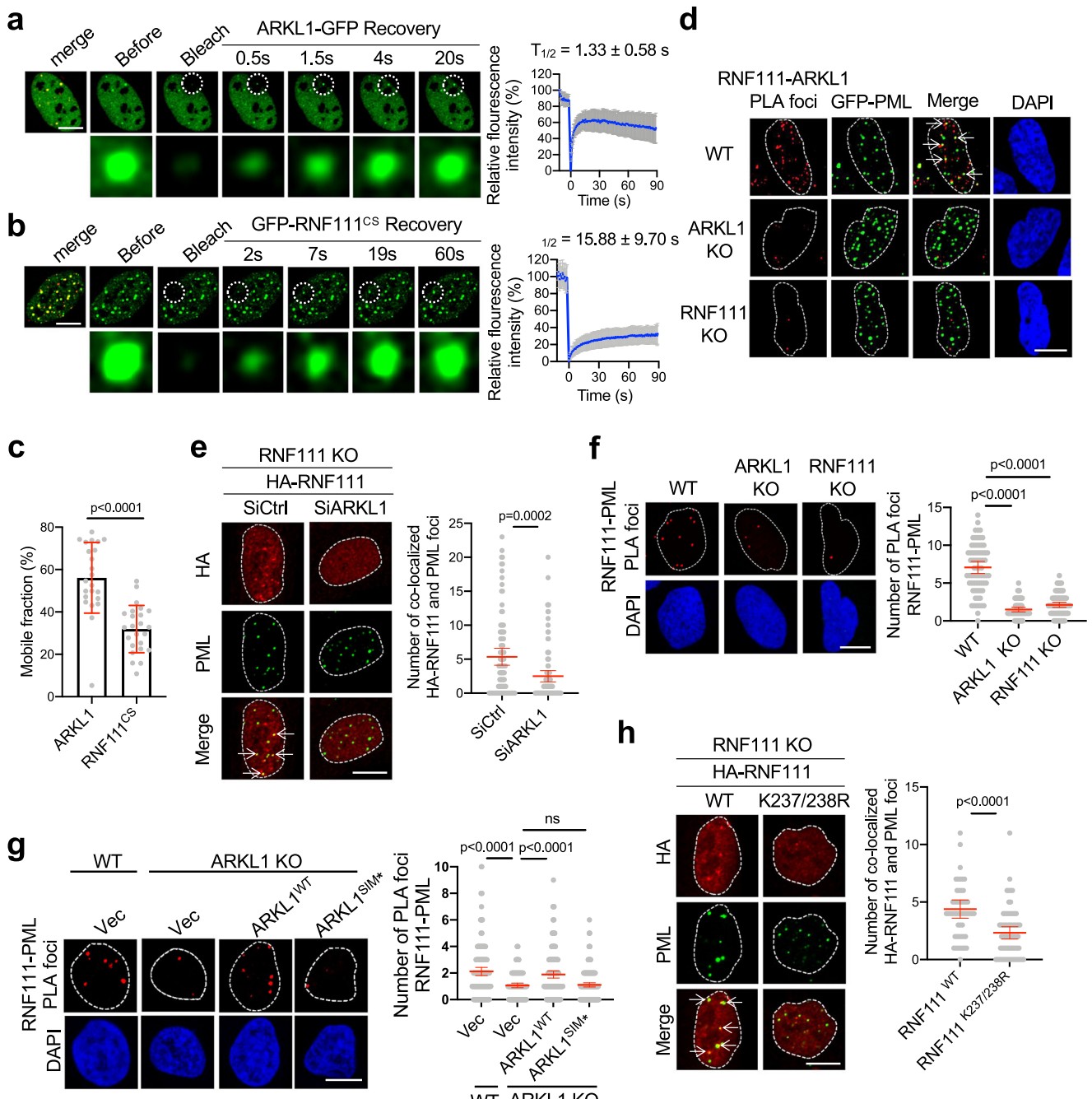

**Fig. 4 | ARKL1 promotes RNF111 localization to PML NBs. a, b** FRAP of ARKL1-GFP and GFP-RNF111$^{CS}$ in PML NBs. Mcherry-PML and ARKL1-GFP or GFP-RNF111$^{CS}$ are co-expressed in U2OS cells. Photobleaching was performed on indicated PML-NB (marked by mcherry-PML in the "merge" image). Representative images of live-cell imaging at indicated times are shown. Relative GFP intensity in the bleached PML NBs before and after bleach was quantified. Exponential fit and recovery half time ($T_{1/2}$) is shown with mean ± SD ($n = 23$ events). **c** Percentage of mobile fraction after photo bleach was quantified with mean ± SD ($n = 23$ events). **d** RNF111-ARKL1 PLA was performed in indicated cells expressing GFP-PML. Arrows point to PLA foci that overlap with PML NBs. **e** Decreased RNF111 localization to PML NBs in ARKL1-deficient cells. Immunofluorescence (IF) was quantified for *RNF111* KO cells complemented with HA-RNF111 and treated with either siCtrl ($n = 98$) or siARKL1 ($n = 100$) from two biological replicates. **f** RNF111 co-localization/interaction with PML is dependent on ARKL1. RNF111-PML PLA was quantified for WT ($n = 71$), *ARKL1* KO ($n = 62$) and *RNF111* KO ($n = 77$) cells from two biological replicates. **g** ARKL1 SIMs are required for RNF111 recruitment to PML NBs. RNF111-PML PLA was quantified for WT U2OS cells expressing vector ($n = 164$) or *ARKL1* KO expressing vector ($n = 133$), ARKL1$^{WT}$ ($n = 171$), ARKL1$^{\Delta S}$ ($n = 200$) from two biological replicates. Expression of the indicated proteins is shown in Supplementary Fig. 4g. **h** RNF111 K237/K238R mutation disrupts RNF111 localization to PML NBs. IF was quantified for *RNF111* KO cells complemented with RNF111$^{WT}$ ($n = 53$) or RNF111$^{K237/238R}$ ($n = 60$) from two biological replicates. Quantification is shown with mean ± 95% CI for **e–h**. Statistics: two-tailed unpaired *t*-test for **c, e, h**, one-way Anova with Sidak's correction for **f, g**. Scale bar, 10 μm for **a, b, d–h**. Source data are provided as a Source Data file.

## CK2 phosphorylates ARKL1 and promotes ARKL1-RNF111 interaction

It has been shown that ARKL1 binds to the regulatory subunit of serine/threonine kinase CK2, CK2β, through a serine-rich stretch[37]. We investigated whether CK2 or ARKL1 interaction with CK2β plays a role in the recruitment of RNF111 to PML NBs and the ubiquitination of BLM. We found that treatment with TBB, an inhibitor of CK2, resulted in a more distributive pattern of RNF111 in the nucleus with decreased foci

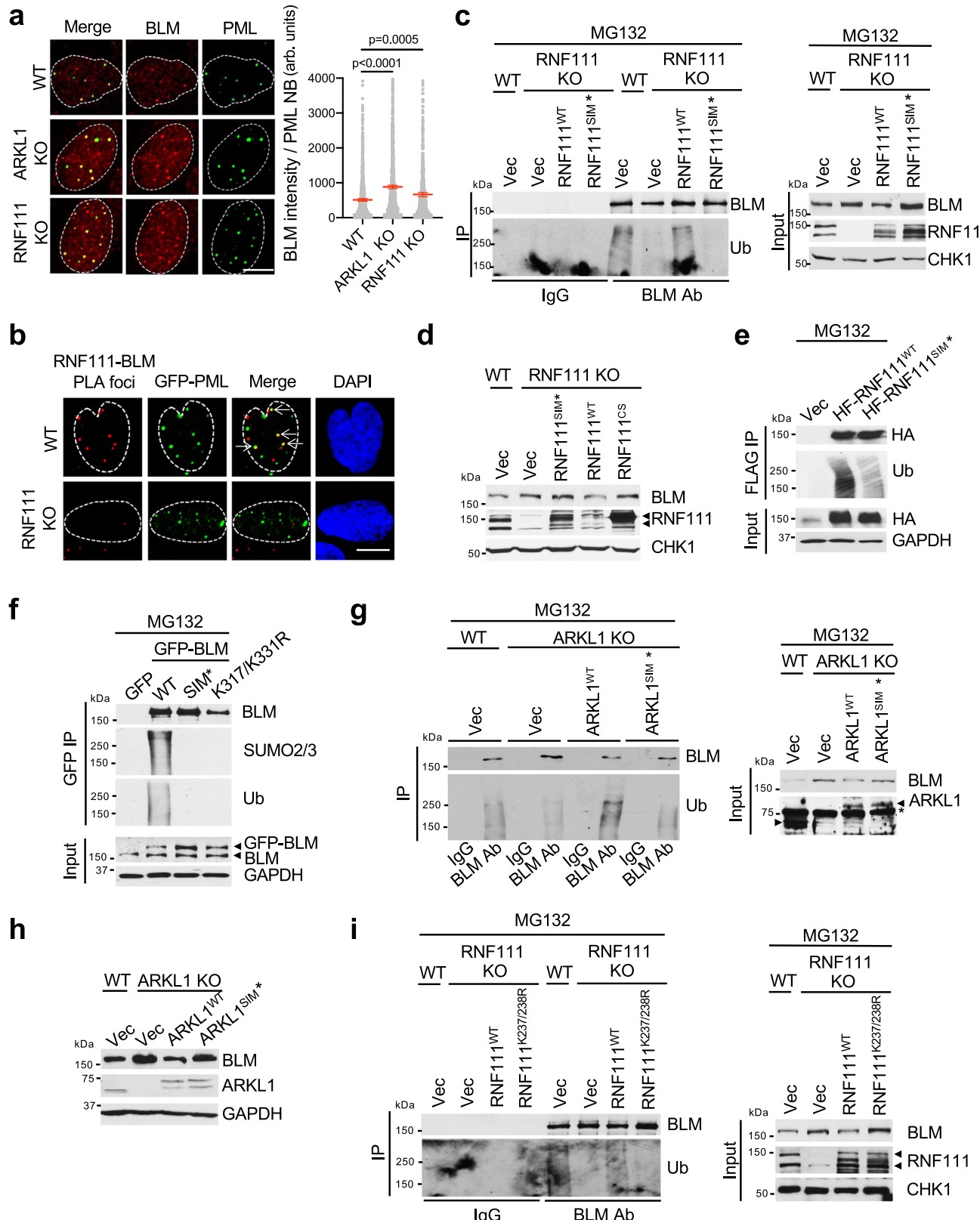

that colocalize with PML NBs (Fig. 6a). The PLA of HA-RNF111 and PML also showed decreased colocalization of RNF111 with PML when the cells were treated with TBB (Fig. 6b). These indicate that CK2 activity promotes RNF111 recruitment to PML NBs. Interestingly, although TBB treatment did not decrease the protein levels of RNF111 or ARKL1 (Supplementary Fig. 6a), it led to reduced BLM ubiquitination, elevated BLM protein levels and increased BLM intensity in PML NBs as those

seen in ARKL1 or RNF111-deficient cell (Fig. 6c, d and Supplementary Fig. 6b). Thus, CK2 plays a role in promoting RNF111 localization to PML NBs and controlling BLM ubiquitination and protein levels.

We then investigated whether ARKL1-CK2β interaction is involved. We used PLA of endogenous RNF111 and PML to examine the localization of RNF111 to PML NBs in *ARKL1* KO cells complemented with either ARKL1[WT] or a serine-rich region deletion mutant of ARKL1

**Fig. 5 | RNF111-mediated ubiquitination of BLM is associated with PML NBs.**
**a** Increased BLM levels in PML NBs in *RNF111* and *ARKL1* KO cells. BLM intensity in each PML NB in immunofluorescence (IF) was quantified with arbitrary (arb.) units with mean ± 95% CI for WT (*n* = 1532), *ARKL1* KO (*n* = 1617), *RNF111* KO (*n* = 952) U2OS cells from two biological replicates. One-way ANOVA with Sidak's correction was used for statistics. **b** Detection of RNF111 and BLM interaction in PML NBs by PLA in WT or *RNF111* KO U2OS cells expressing GFP-PML. RNF111-BLM PLA foci that colocalize with PML NBs are indicated by arrows. **c** RNF111 SIM domains are required for BLM ubiquitination. The IPs were performed under denature condition using lysates from WT or *RNF111* KO U2OS cells expressing indicated constructs and treated with MG132. CHK1 was used as a loading control. **d** RNF111^SIM* mutant fails to reduce the elevated BLM levels in *RNF111* KO cells. **e** RNF111 SIM domains are

required for self-ubiquitination. 293T cells expressing indicated constructs were used. **f** BLM ubiquitination is dependent on its SIMs and SUMOylation at K317/331. GFP IP was performed under denature condition using lysates of 293T cells expressing GFP or GFP-tagged BLM constructs. **g** ARKL1 SIMs are required for BLM ubiquitination. WT or *ARKL1* KO Hela cells expressing indicated constructs were used. In the Input panel, "*" non-specific band; left arrow, endogenous ARKL1; right arrow, Flag-tagged ARKL1. **h** Expression of WT but not SIM mutant of ARKL1 reduced the elevated BLM levels in *ARKL1* KO cells. **i** RNF111 SUMOylation sites K237/K238 are required for BLM ubiquitination. WT or *RNF111* KO U2OS cells expressing indicated constructs were used. Scale bar, 10 µm for **a**, **b**. Source data are provided as a Source Data file.

(ARKL1^ΔS) that lacks the interaction with CK2β[37]. *ARKL1* KO cells complemented with ARKL1^ΔS showed a reduced number of PLA foci when compared to that of ARKL1^WT-complemented cells, indicative of a deficiency in RNF111 recruitment to PML NBs (Fig. 6e and Supplementary Fig. 6c). In addition, the elevated BLM intensity in PML NBs in *ARKL1* KO cells is reduced in cells complemented with ARKL1^WT but not ARKL1^ΔS or ARKL1^SIM* (Fig. 6f and Supplementary Fig. 6d). These indicate that, like the role of ARKL1-RNF111 interaction, ARKL1-CK2β interaction is critical for a role of ARKL1 in limiting BLM levels.

We then investigated how CK2 plays a role in RNF111-mediated BLM ubiquitination and protein degradation in PML NBs. We first investigated whether ARKL1 is a substrate of CK2 as CK2β has a role in recruiting substrates of CK2[39]. We found that immunoprecipitated ARKL1 can be detected by anti- phosphor-CK2 substrate (pS/pTDXE) antibody and inhibition of CK2 by TBB abolished the detection, supporting that ARKL1 is phosphorylated by CK2 and inhibition of CK2 decreases the phosphorylation (Fig. 6g). In addition, HA-ARKL1 WT but not ΔS mutant can be detected in the immunoprecipitates from anti-phosphor-CK2 substrate (pS/pTDXE) antibody, indicating that the interaction of ARKL1 with CK2β is critical for the phosphorylation of ARKL1 by CK2 (Fig. 6h). Accordingly, knocking down CK2β abolished the phosphorylation of ARKL1 by CK2 (Supplementary Fig. 6e). It also leads to reduced BLM ubiquitination and elevated BLM protein levels (Supplementary Fig. 6f, g). Together, these data support the idea that ARKL1 interaction with CK2β leads to a phosphorylation of ARKL1 by CK2.

In the search of potential CK2 phosphorylation site with a consensus CK2 phosphorylation motif S/T-x-x-D/E, we noticed that serine 328 (S328) residue resides in a CK2 recognition motif nearby the first SIM domain of ARKL1 (Fig. 6i and Supplementary Figs. 3c and 6h). CK2-mediated phosphorylation of serine residue in an acidic patch adjacent to a SIM has been identified in several SIM-containing proteins, such as PIAS1, PML, PMSCL1, RAP80 and IE2, playing a role in promoting SIM-SUMO binding (Supplementary Fig. 6i)[40–42]. We first generated an ARKL1 mutant with four serine residues (S327, S328, S330, and S385) near the two SIM domains mutated to alanine (4A:S327/328/330/385A) and found that 4 A mutant abolished CK2 phosphorylation that can be recognized by anti- phosphor-CK2 substrate (pS/pTDXE) antibody (Fig. 6i). Mutation of S328 in the CK2 recognition motif alone also lost CK2 phosphorylation, indicating that S328 is a CK2 phosphorylation site (Fig. 6i). Sequence surrounding S328 and the CK2 substrate motif and the nearby SIM domain of ARKL1 is conserved though various species (Supplementary Fig. 6h).

We then investigated whether the phosphorylation of ARKL1 by CK2 is important for the SIM-SUMO binding of ARKL1 and RNF111. We found that inhibition of CK2 by TBB reduced the binding of ARKL1 and RNF111, suggesting a role of CK2 phosphorylation in promoting ARKL1-RNF111 interaction (Fig. 6j). Deletion of CK2 binding region (ARKL1^ΔS) or mutation of CK2 phosphorylation site (ARKL1^S328A) reduced the binding of ARKL1 and RNF111 (Fig. 6k). Phosphomimic mutation of the serine residues nearby the SIM domains (ARKL1^S327/328/330/385D), however, still retains the interaction of ARKL1 and RNF111, consistent with the

role of phosphorylation in promoting ARKL1-RNF111 interaction (Supplementary Fig. 6j).

Together, these data indicate that CK2 phosphorylation of a serine residue S328 next to the first SIM domain of ARKL1 is critical for the SIMs of ARKL1 binding to RNF111 for the recruitment of RNF111 to PML NBs and regulation of BLM ubiquitination and protein levels.

## ARKL1-RNF111-mediated control of BLM levels maintains G4 levels

BLM plays an important role in unwinding G4 DNA structures and suppressing cellular G4 levels[3–5,9,43]. We asked whether ARKL1-RNF111-mediated control of BLM level is involved in G4 regulation. Immunofluorescence showed that treatment with Pyridostatin (PDS), a G4 stabilizer, significantly increased G4 levels that can be detected by an anti-G4 antibody, BG4[44]. (Supplementary Fig. 7a). We found that G4 levels in the nucleus were reduced in *RNF111* KO cells (Fig. 7a, b). Depletion of BLM in *RNF111* KO cells restored G4 intensity to the control level, indicating that the effect of loss of RNF111 on G4 levels is dependent on BLM (Fig. 7a and Supplementary Fig. 7b). Indeed, G4 intensity is highly dependent on BLM protein levels, BLM over-expression greatly decreased G4 intensity and treatment of a BLM helicase inhibitor ML216 rescued the reduction of G4 intensity caused by BLM overexpression (Supplementary Fig. 7c). These are consistent with the previous findings that BLM resolves G4 structure through its helicase activity[5,9]. Treatment with ML216 also restored the G4 levels in *RNF111* KO cells, further supporting the idea that BLM mediates the role of RNF111 in regulating G4 levels (Fig. 7b).

We then assessed the importance of RNF111-ARKL1-mediated ubiquitination and control of BLM levels in PML NBs in the regulation of G4 levels. We found that the reduced G4 intensity in *RNF111* KO cells can be rescued by complementation with RNF111^WT but not empty vector, catalytic inactive RNF111^CS or RNF111^SIM* (Fig. 7c). This is consistent with the role of RNF111 E3 ligase activity and PML NBs localization in the regulation of BLM levels. In addition, consistent with the role of ARKL1 in promoting RNF111-mediated degradation of BLM, loss of ARKL1 led to reduced levels of G4 and depletion of BLM or treatment with ML216 restored the G4 levels in ARKL1-deficient cells (Fig. 7d and Supplementary Fig. 7d, e). As a control, depletion of RNF111 or ARKL1 did not result in excessive DNA damage since γH2AX levels was not elevated in RNF111 or ARKL1 depleted cells (Supplementary Fig. 7f). The reduced nuclear G4 levels were also observed in BJ cells depleted of ARKL1 or RNF111, or when staining was performed with an additional G4 antibody (1H6)[45] (Supplementary Fig. 7g–i). Furthermore, due to the role of CK2 in promoting ARKL1-RNF111-mediated degradation of BLM, inhibition of CK2 with TBB also led to a decrease in G4 levels (Fig. 7e). Together, these data demonstrate that CK2- and ARKL1-RNF111-dependent control of BLM levels play a critical role in regulating G4 levels in unperturbed cells.

## Discussion

Our study reveals a regulatory mechanism that controls BLM protein levels involving SUMO-SIM mediated RNF111-ARKL1 interaction, CK2

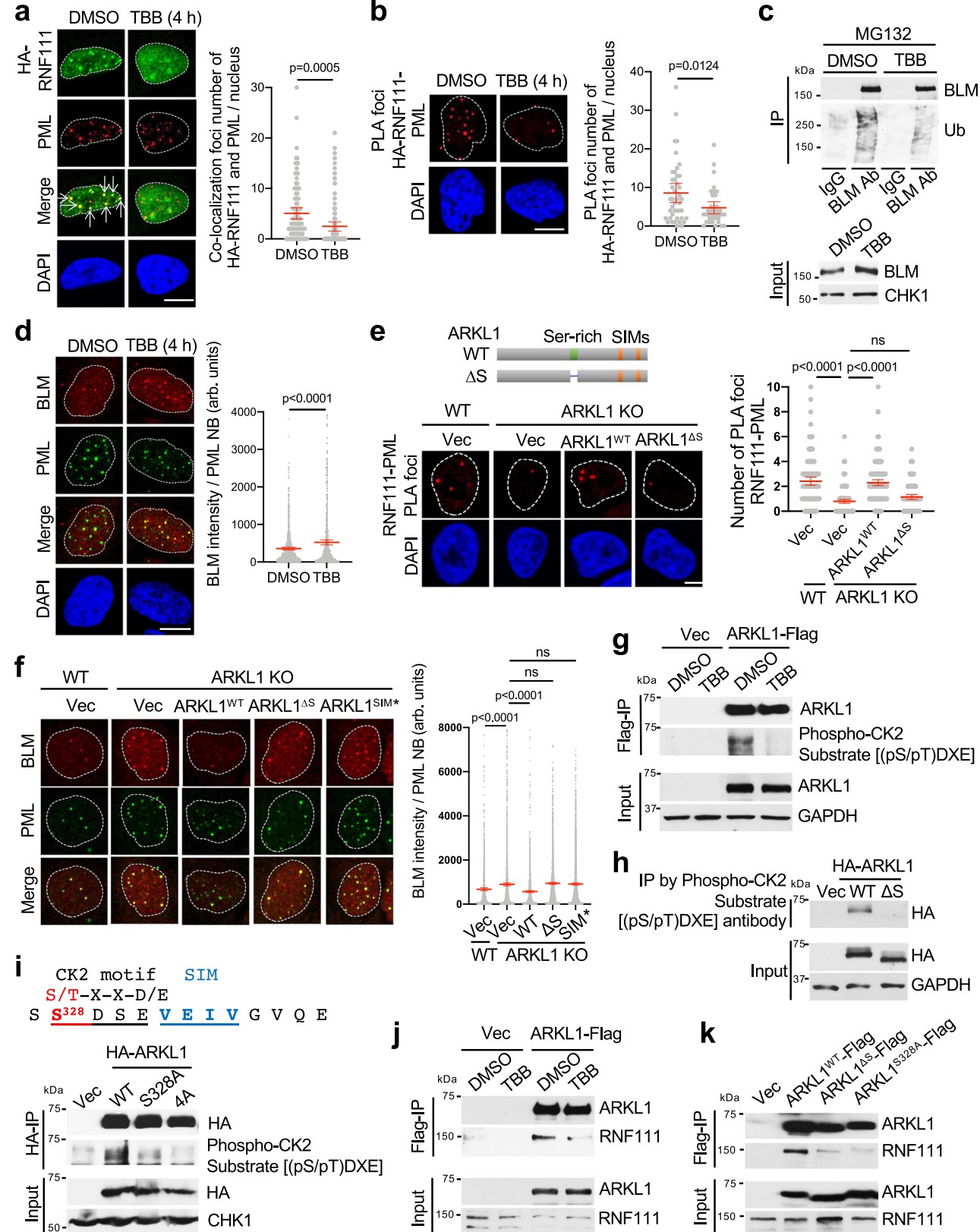

activity and PML NBs and demonstrates that RNF111-ARKL1-dependent ubiquitination of BLM in PML NBs is critical for limiting BLM levels in the regulation of G4 in the nucleus (Fig. 7f).

STUBLs are a group of ubiquitin E3 ligases that possess both SIMs and RING domain that selectively target SUMOylated proteins[20–23,46]. A canonical STUBL utilizes its SIMs to bind SUMOylated substrate for the ubiquitination of the substrate. In RNF111-mediated ubiquitination of BLM, however, the interaction of RNF111 and BLM is independent of the SIMs of RNF111 and the SUMOylation of BLM (Fig. 1). In addition, it requires ARKL1, a N-terminal paralog of RNF111 which contains SIMs but lacks RING domain (Fig. 2). ARKL1 interacts with RNF111 in a SIM-SUMO-dependent manner via the SIMs of ARKL1 and the SUMOylation

**Fig. 6 | CK2 phosphorylates ARKL1 promoting ARKL1-RNF111 interaction and BLM ubiquitination. a** CK2 inhibitor TBB treatment abolishes HA-RNF111 localization to PML NBs. Immunofluorescence (IF) in *RNF111* KO U2OS cells expressing HA-RNF111 was quantified for DMSO-treated (*n* = 101) or TBB (50 μM, 4 h)-treated (*n* = 100) cells from two biological replicates. **b** PLA in *RNF111* KO cells expressing HA-RNF111 was quantified for DMSO-treated (*n* = 43) or TBB-treated (*n* = 37) cells. **c** TBB treatment reduces BLM ubiquitination. **d** TBB treatment leads to increased BLM levels in PML NBs. IF was quantified with arbitrary (arb.) units for DMSO-treated (*n* = 1354) or TBB-treated (*n* = 903) cells. **e** The serine-rich region of ARKL1 is required for promoting RNF111-PML colocalization. PLA was quantified for Hela WT expressing vector (*n* = 152) or *ARKL1* KO expressing vector (*n* = 178), ARKL1$^{WT}$ (*n* = 180), or ARKL1$^{ΔS}$ (*n* = 130) from two biological replicates. Expression of the indicated proteins is shown in Supplementary Fig. 6c. **f** ARKL1$^{WT}$ but not ARKL1$^{ΔS}$ or ARKL1$^{SIM*}$ mutant reduces the elevated BLM levels in PML NBs in *ARKL1* KO cells.

Quantifications are shown for U2OS WT expressing vector (*n* = 1302) or *ARKL1* KO cells expressing vector (*n* = 2109), ARKL1$^{WT}$ (*n* = 1467), ARKL1$^{ΔS}$ (*n* = 2307), ARKL1$^{SIM}$ (*n* = 3015). Expression of the indicated proteins is shown in Supplementary Fig. 6d. **g** TBB treatment leads to decreased CK2 phosphorylation of ARKL1. The IPs were performed under denature condition. **h** ARKL1$^{ΔS}$ mutant cannot be phosphorylated by CK2. Hela cells expressing indicated constructs were used. **i** Identification of CK2 phosphorylation site on ARKL1. 293T cells expressing vector, HA-tagged ARKL1 WT, S328A or 4 A (S327/328/330/385 A) mutant were used. **j** TBB treatment reduces the interaction of ARKL1-Flag with RNF111. **k** ARKL1$^{ΔS}$ and ARKL1$^{S328A}$ mutants exhibit decreased interaction with RNF111. Quantifications are shown with mean ± 95% CI for **a**, **b**, **d**–**f**. Statistics: two-tailed unpaired t-test for **a**–**d**, one-way ANOVA with Sidak's correction for **e**, **f**. Scale bar, 10 μm for **a**, **b**, **d**–**f**. Source data are provided as a Source Data file.

of RNF111 at K237/238 residues (Fig. 3). Moreover, the SIM-SUMO interaction of ARKL1 and RNF111 is largely dependent on CK2 activity (Fig. 6). Through binding to CK2β, ARKL1 is phosphorylated by CK2 on the residue of S328 which is adjacent to the first SIM of ARKL1. The phosphorylation greatly enhances the binding of ARKL1 SIMs to SUMOylated RNF111. The RNF111-medicated BLM ubiquitination also involves PML NBs. Localization of both RNF111 and BLM to PML NBs are required for efficient ubiquitination of BLM (Fig. 5). Thus, the RNF111-ARKL1 noncanonical STUBL activity is distinct from the canonical STUBL in the following aspects: (1) it requires a facilitator protein (ARKL1); (2) the SIM-SUMO interaction is NOT involved in the interaction of the E3 ligase (RNF111) and the substrate (BLM) but is involved in the interaction of the E3 ligase (RNF111) with the facilitator (ARKL1); (3) it is subjected to the regulation of CK2 that promotes SIM-SUMO interaction; (4) it involves PML NBs in which the localization of the E3 ligase and the substrate is essential for the ubiquitination.

BLM is primarily associated with PML NBs in unperturbed cells[11–13]. Our study indicates that PML NB serve as a hub for RNF111-mediated ubiquitination and degradation of BLM (Figs. 4, 5). BLM mutants deficient in PML NBs localization (BLM$^{K317/331R}$ and BLM$^{SIM*}$) fail to be ubiquitinated. It is noteworthy that both mutants are intact in their interaction with RNF111 in the in vitro co-immunoprecipitation experiment (Fig. 1h), which further highlights the importance of PML NBs in BLM ubiquitination. Similarly, the localization of RNF111 to PML NBs is also critical for BLM ubiquitination. RNF111$^{SIM*}$ with a deficiency in localizing to PML NB is compromised in its ability to mediate BLM ubiquitination. Due to the role of ARKL1-RNF111 interaction in promoting RNF111 to PML NBs, ARKL1$^{SIM*}$ and RNF111$^{K237/238R}$ SUMOylation mutants which abolish the interaction of ARKL1 and RNF111 are defective in mediating BLM ubiquitination. Thus, RNF111-mediate BLM ubiquitination occurs only efficiently in PML NBs.

PML-NBs may promote the efficiency of RNF111-mediated BLM ubiquitination by concentrating the E3 ligase RNF111, the substrate BLM and other related proteins through compartmentalization of these proteins, promoting their interaction and facilitating the ubiquitination process. Consistent with this idea, several previous studies have reported the presence of components of the ubiquitin/proteasome pathway associated with PML NBs[47–50]. Thus, it is possible that PML-NB provides a catalytic platform for RNF111-mediated ubiquitination of BLM that would otherwise proceed less efficiently in the absence of their localization to PML NBs. In addition, PML NBs may provide a selective environment that allows RNF111 catalytic activity for proteins that are SUMOylated. This is consistent with previous suggestions that PML NBs may be sites of degradation of some SUMOylated proteins[47,48,51,52]. SUMOylation and SUMO-SIM interaction play essential roles during the formation of PML NBs and the recruitment of proteins to PML NBs[14,15,18,53]. The SUMO-related regulation in PML NBs likely also involves the CK2 activity. In the regulation of BLM ubiquitination, CK2 phosphorylates S328 of ARKL1 adjacent to the first SIM of ARKL1 promoting ARKL1 binding to SUMOylated RNF111. CK2

phosphorylation of a serine residue nearby the SIM domain also has been shown facilitating SIM-SUMO interaction in the regulation of several PML NBs associated proteins such as DAXX, PIAS1 and cytomegalovirus transactivator IE2[40–42,54–56]. Thus, CK2 signaling likely plays a critical role in the SUMO-related control of protein stability in PML NBs. Finally, PML NBs might also serve as a hub to spatially limit the catalytic activity of RNF111. The sequestering of the protein degradation machinery to PML NBs can provide an advantageous edge that restrains uncontrolled ubiquitination and degradation of BLM or prevents access of unwarranted regulation of RNF111 catalytic activity. Therefore, PML NBs may play an important role in promoting efficiency, selectivity and providing spatial restraint for RNF111-mediated BLM ubiquitination and control of protein stability. Many proteins are associated with PML NBs, it is possible that RNF111-mediated BLM ubiquitination in PML NBs represents a universal mechanism for the regulation of these proteins.

Earlier studies have shown that BLM protein level is low in normal human tissues and high-level expression is restricted to the proliferating compartment of the tissue and tumors, indicating that BLM level is tightly regulated in cells[57]. Our study indicates that ARKL1-RNF111-dependent ubiquitination of BLM in PML NBs serves as a regulatory mechanism in controlling BLM protein stability thereby fine-tuning the nucleoplasmic pool of BLM levels in unperturbed cells. Since BLM helicase is highly active in binding and unwinding G4 structure, the regulation of the amount of BLM in cells likely impacts on G4 levels[3,5,10]. Our data indicate that RNF111-ARKL1-dependent limit of BLM levels is critical in maintaining G4 levels in unperturbed cells (Fig. 7). Sequences that form G4 are widely present in the genome. It has been indicated that while G4 can present as an obstacle to DNA processing and is related to genome instability, it also forms naturally under physiological condition playing regulatory roles in DNA transcription, transcription, translation and telomere biology[6–8]. Although the functional importance of maintenance of G4 levels remains to be explored in the future, our study provides an aspect of the regulation of G4 through the control of BLM protein levels.

Due to the roles of BLM in multiple aspects of DNA processing such as DNA replication, repair, recombination, and telomere maintenance, the control of BLM protein level may also be critical in other aspects of the function of BLM in maintaining DNA metabolism homeostasis in cells. G4 structures are enriched in telomere since telomeric DNA consists of tandem copies of short G-rich repeats (5′-TTAGGG-3′). BLM also has been reported to promote telomere replication by resolving G4 structures[10]. In addition, in cancer cells where an alternative lengthening of telomere (ALT) mechanism is active, BLM is associated with the ALT-associated PML NBs (APBs) and plays a critical role in ALT[58]. The regulation of BLM through RNF111-ARKL1-dependent control may therefore also contributes to telomere maintenance and ALT.

Together, our study identifies a RNF111 and ARKL1 involved non-canonical STUBL activity for the ubiquitination and degradation of

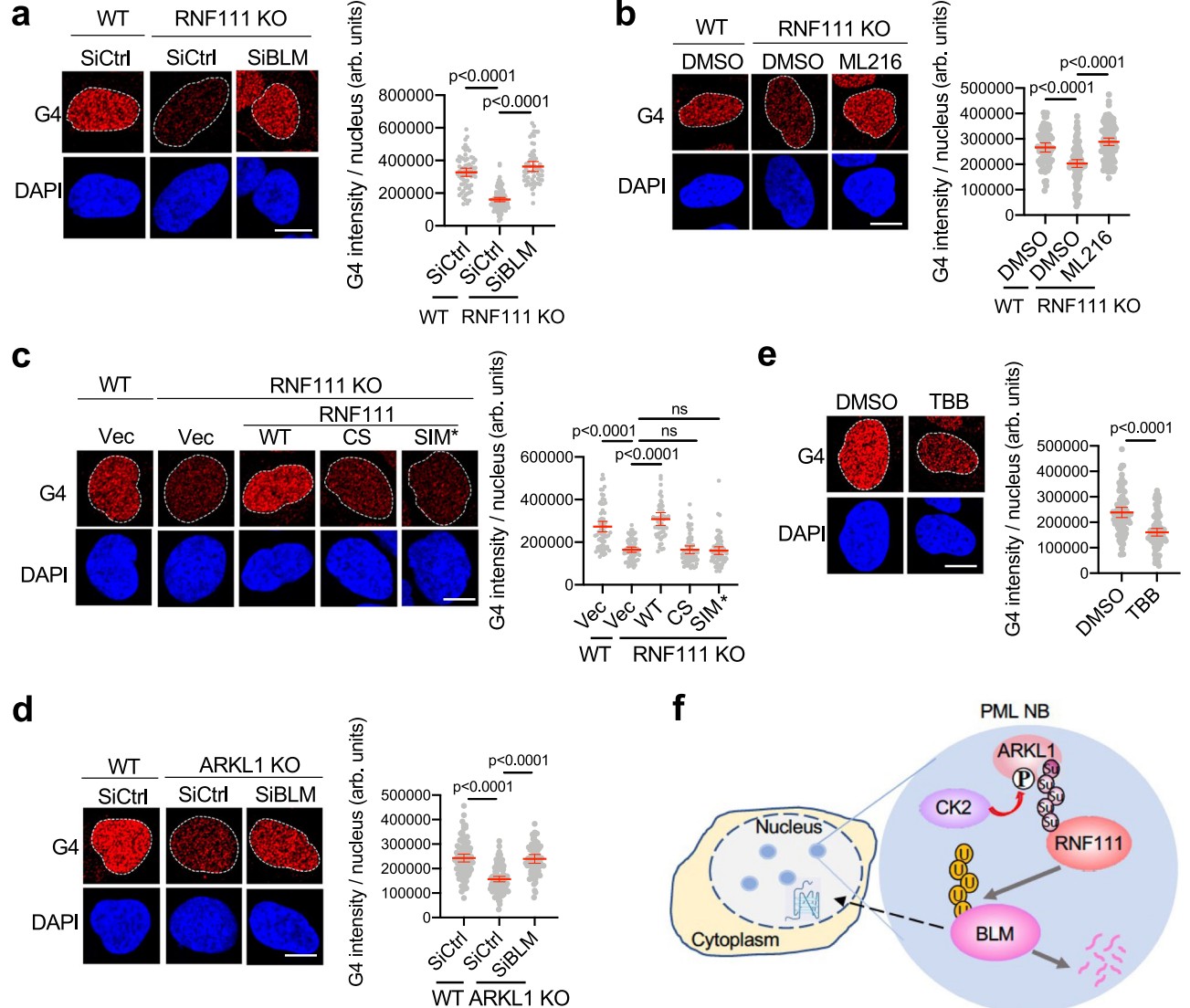

**Fig. 7 | RNF111-ARKL1 and CK2 regulate G-quadruplex in the nucleus through modulating BLM levels. a** Decreased G4 levels in *RNF111* KO nucleus can be rescued by BLM knockdown. Data from two biological replicates are quantified with arbitrary (arb.) units for U2OS WT treated with siCtrl (*n* = 68), *RNF111* KO treated with siCtrl (*n* = 101) or siBLM (*n* = 57). Western blots are shown in Supplementary Fig. 7b. **b** Decreased G4 levels in *RNF111* KO cells are rescued by treatment with BLM helicase inhibitor ML216 (12.5 μM, 72 h). Data from two biological replicates are quantified for WT cells treated with DMSO (*n* = 66), *RNF111* KO treated with DMSO (*n* = 95) or ML216 (*n* = 94). **c** Decreased G4 intensity in *RNF111* KO cells is rescued by expression of RNF111^WT (*n* = 56), but not empty vector (*n* = 68), RNF111^CS (*n* = 61), or RNF111^SIM* (*n* = 62) from two biological replicates. **d** Decreased G4 intensity in *ARKL1*

KO U2OS nucleus can be rescued by BLM knockdown. Western blots are shown in Supplementary Fig. 7d. Data of two biological replicates are quantified for WT cells treated with siCtrl (*n* = 85), ARKL1 KO cells treated with siCtrl (*n* = 98) or SiBLM (*n* = 68). **e** CK2 inhibitor TBB treatment leads to decreased G4 intensity in the nucleus. Data of two biological replicates are quantified for U2OS cells treated with DMSO (*n* = 87) or TBB (50 μM, 4 h) (*n* = 86). **f** A proposed model for ARKL1-RNF111-dependent ubiquitination of BLM in PML NBs limiting BLM protein level for the regulation of G4 in the nucleus. Quantifications are shown with mean ± 95% CI. Statistics: one-way ANOVA with Sidak's correction for **a**–**d** and two-tailed unpaired t-test for **e**. Scale bar, 10 μm for **a**–**e**. Source data are provided as a Source Data file.

BLM and proposes a model that BLM level in the cell is limited by RNF111-ARKL1-mediated ubiquitination and degradation in PML NBs under the regulation of CK2 activity for maintaining G4 levels.

## Methods

### Cell line and cell culture

U2OS cells (ATCC, Cat# HTB-96) were grown in McCoy's 5A with L-glutamine medium (Corning, Cat# 10-050-CV) with 10% fetal bovine serum (Gibco, Cat# 10437-028). 293 T (ATCC, Cat# CRL-3216), Hela (ATCC, Cat# CCL-2), and BJ cells (ATCC, Cat# CRL-2522) were grown in Dulbecco's modified Eagle's medium (DMEM) (Corning, Cat# 10-013-CV) with 10% fetal bovine serum. For SiRNA transfection, cells were seeded one day before transfection. Lipofectamine™ RNAiMAX

Transfection Reagent (Thermo Fisher, Cat# 13778150) was used for transfection based on manufacture's protocol. For plasmid transfection, Lipofectamine™ 2000 Transfection Reagent (Thermo Fisher, Cat# 11668019) was used based on manufacture's protocol. RNF111 KO U2OS was generated by CRISPR-Cas9 using Invitrogen TrueGuide™ sgRNA (Thermo Fisher, Cat# A35533) and TrueCut™ Cas9 Protein v2 (Thermo Fisher, Cat# A36497), Lipofectamine™ CRISPRMAX™ Cas9 Transfection Reagent (Thermo Fisher, Cat# CMAX00003) was used for transfection according to manufacture's protocol. ARKL1 KO U2OS and Hela cells were generated by CRISPR-Cas9 using pLentiCRISPR lentivirus carrying Cas9 and sgRNA targeting ARKL1 sequence 5'-ACAGAATGGCCGAGTCGCCA-3'. Cells were subjected to puromycin selection and single colonies were selected and expanded for a screen

of ARKL1 clones. For Cycloheximide-chase assay, cells were treated with 100 µg/ml cycloheximide (Sigma-Aldrich, C7698).

### SiRNAs, plasmids, and antibodies

SiRNAs used in this study are listed in Supplementary Table 1. The plasmids generated in this study are listed in Supplementary Table 2. GFP-BLM plasmid was a gift from Nathan Ellis (Addgene plasmid #80070)[59]. pAID5.3-C was a gift from Masato Kanemaki (Addgene plasmid 145813)[60]. pCMV-Tol2 plasmid was a gift from Stephen Ekker (Addgene plasmid#31823)[61]. The primers used in this study for generating various mutants for BLM, ARKL1, and RNF111 are listed in Supplementary Table 3. The antibodies and their dilution in the applications used in this study are listed in Supplementary Table 4.

### Immunoblotting and immunoprecipitation

For cell lysis, cells were harvest from plates by scrapers and immediately lysed by RIPA lysis buffer with 1× protease inhibitor cocktail, 1 mM PMSF, 50 mM β-Glycerophosphate, 5 mM NaF, 1 mM $Na_3VO_4$ and 20 mM N-ethylmaleimide. The lysate was sonicated in ice water and centrifuged at $21,000 \times g$ for 10 min at 4 °C. The supernatant was collected and 5× SDS-PAGE sample loading buffer was added. Samples were denatured at 95 °C for 5 min. For immunoprecipitation, cells were harvest from plates by scrapers and immediately lysed by modified NETN lysis buffer (20 mM Tris-HCl pH 7.6, 95 mM NaCl, 1 mM EDTA, 0.5% Nonidet P-40, 1× protease inhibitor cocktail, 1 mM PMSF, 50 mM β-Glycerophosphate, 5 mM NaF, 1 mM $Na_3VO_4$ and 20 mM N-ethylmaleimide). The lysate was sonicated in ice water and centrifuged at $21,000 \times g$ for 10 min at 4 °C. The supernatant was collected and Flag/HA beads or antibodies conjugated with Protein A/G beads were added. After incubation at 4 °C overnight, beads were washed by modified NETN buffer for 4 × 15 min. Samples were eluted by 2× SDS-PAGE sample loading buffer at 95 °C for 5 min. For Immunoprecipitation to detect protein ubiquitination or SUMOylation, cells were treated with 10 µM MG132 (Sigma, Cat# 474790) for 4 hrs and lysed by modified NETN lysis buffer with 0.4% SDS. For SUMOylation, similar results were obtained when cells were not treated with MG132. The lysate was sonicated in ice water and centrifuged at $21,000 \times g$ for 10 min at 4 °C. The supernatant was collected and SDS was diluted to 0.1% by modified NETN buffer. Flag/HA beads or antibodies conjugated with Protein A/G beads were added. After incubation at 4 °C overnight, beads were washed by NETN buffer for 4 × 15 min. Samples were eluted by 2× SDS-PAGE sample loading buffer at 95 °C for 5 min.

### In vitro ubiquitination assay

RNF111[WT]-Flag or RNF111[CS]-Flag expressed in 293 T cells was immunoprecipitated with Flag beads and eluted with 200 ng/µL Flag peptides (Thermo Fisher, Cat# A36805). Eluted RNF111[WT]-Flag was then applied to Zeba™ Spin Desalting Columns (Thermo Fisher, Cat# 89890) for further purification. GFP-BLM was purified from 293 T cells expressing GFP-BLM using GFP antibody-couple protein A/G beads. The in vitro purification reaction was performed following the instruction of Enzo Ubiquitinylation kit (Enzo, Cat# BML-UW9920). Briefly, purified GFP-BLM and RNF111-Flag were incubated with 1x ubiquitination buffer (Enzo, Cat# BML-KW9885-0005) with 2.5uM ubiquitin (Enzo, Cat# BML-UW8705-0100), 100 nM ubiquitin Activating Enzyme E1 (R&D Systems, Cat# E-304-050), 2.5 µM ubiquitin-conjugating enzyme E2 UbcH5c (Enzo, Cat# BML-UW9070-0100), 5 mM $Mg^{2+}$-ATP (Enzo, Cat# BML-EW9805-0100), 1 mM DTT, 20 U/mL pyrophosphatase (Sigma, Cat# I1643) at 37 °C with rotation for 2 h. Reaction was quenched by 2x non-reducing gel loading buffer.

### Immunofluorescence

Cells grown on coverslips were fixed with 4% paraformaldehyde in phosphate buffered saline (PBS) at room temperature for 10 min, followed by PBS washing for 4 × 5 min. Cells were then permeabilized with permeabilization buffer (25 mM HEPES pH 7.4, 50 mM NaCl, 1 mM EDTA, 3 mM $MgCl_2$, 300 mM Sucrose, 0.5% Triton-X) at 4 °C for 6 min, followed by PBS washing for 4 × 5 min. After permeabilization, cells were incubated with primary antibodies in immunofluorescence buffer (3% bovine serum albumin (BSA) and 0.2% Triton-X in PBS) at 4 °C overnight, followed by 4 × 5 min PBS washing. After incubation with dye-conjugated secondary antibodies (1:1000 dilution) for 1 h at 37 °C and PBS washing for 4 × 5 min, cells were mounted using ProLong™ Gold Antifade Mountant with DAPI (Thermo Fisher, Cat# P36931). Images were collected by Nikon A1-Confocal using ×60 objective and analyzed by NIH Elements AR software (AR 5.21.03 64 bit).

### Fluorescence recovery after photobleaching and live-cell imaging

FRAP acquisition was performed with A1-Nikon confocal located at the department of Genetics MD Anderson (BSRB Microscopy facility) equipped with LUNV laser launch and DU-G GaASP detector unit and a Nikon Plan-Apo 60X oil 1.4 NA objective. The entire experiments were performed under incubation (temperature, $CO_2$, and humidity control). A baseline of 20 pre-bleach images was initially collected, followed by a bleaching sequence using 25% of our current laser power in the 488 nm laser line and a small circular region of interest (1.3 µm). A total of 180 image frames was recorded for post-bleach recovery continuously until signal reached a steady state. Movies were analyzed using NIS-Elements, and curve fitting was done with GraphPad Prism software. For curve fitting, a single exponential function ($f(t) = \alpha(1-e^{-kT})$, where $T_{1/2}$ (half-time of recovery) is ln 0.5/(-k), and α is the mobile fraction was used. Raw recovery curves were corrected for background and photofading. The lowest fluorescence signal and the time-point after bleaching were scaled to 0, and curves were normalized to 1 based on the reference signal before bleaching.

### Proximity ligation assay

Cells grown on coverslips were fixed with 4% paraformaldehyde in PBS at room temperature for 10 min, followed by PBS washing for 4 × 5 min. Cells were then permeabilized with permeabilization buffer at 4 °C for 6 min, followed by PBS washing for 4 × 5 min. PLA was performed with Duolink reagents (Sigma Aldrich) according to the manufacturer's protocol. Briefly, coverslips were blocked with Duolink® blocking solution at 37 °C for 60 min and incubated with primary antibodies in Duolink antibody diluent at 4 °C overnight, followed by wash buffer A (Sigma, Cat# DUO82049) washing for 2 ×5 min at room temperature. Ligation was performed with ligase in ligation buffer at 37 °C for 60 min, followed by wash buffer A washing for 2 ×5 min at room temperature. Amplification was performed with polymerase in amplification buffer at 37 °C for 100 min, followed by wash buffer B washing for 2 ×10 min at room temperature. After washing with 0.01x Wash Buffer B for 1 min, coverslips were mounted using ProLong™ Gold Antifade Mountant with DAPI. Images were collected by Nikon A1-Confocal using ×60 objective and analyzed by NIH Elements AR software (AR 5.21.03 64 bit).

### Statistics and reproducibility

Statistical analysis was performed with GraphPad Prism 9 (for macOS, version 9.0.0 (86)). One-way ANOVA with Sidak's correction, two-way ANOVA, or two-tailed unpaired t-test was used for statistical analyses as indicated. $P < 0.05$ was considered statistically significant. All experiments were performed with at least three biological repeats except indicated in the figure legends. No statistical method was used to predetermine sample sizes. No data were excluded from the analyses. The experiments were not randomized. The investigators were not blinded to allocation during experiments and outcome assessment. Western blots and microscopic imaging are representative of at least three biological repeats unless specified in the legends.

**Reporting summary**

Further information on research design is available in the Nature Portfolio Reporting Summary linked to this article.

## Data availability

All data supporting the findings of this study are available within the paper and its supplementary information. All data are available from the corresponding author upon request. Source data are provided with this paper.

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

## Acknowledgements
We thank Drs. Nathan Ellis (Memorial Sloan-Kettering Cancer Center) and Stephen Ekker (Institute of Human Genetics of University of Minnesota) for BLM and related plasmids. S.L., E.A., and B.W. were partially supported by NIH grant (CA155025, CA248088). E.A. is a Graduate Scholar in the CPRIT Training Program (RP210028). Use of the Nikon A1 microscope was made possible via a NIH shared Instrumentation Grant (grant no. 1S10OD024976-01 to the MDACC Department of Genetics).

## Author contributions
S.L. and B.W. designed the study, S.L. and B.W. wrote the manuscript. B.W. supervised the project. S.L performed majority of the experiments; E.A. generated ARKL1 KO cells and performed experiments on the binding of ARKL1 to BLM; A.P. performed FRAP and data analyses with S.L.

## Competing interests
The authors declare no competing interests.
