## [Peer Review File · Nature Communications]

A CK2 and SUMO-dependent, PML NB-involved regulatory mechanism controlling BLM ubiquitination and G quadruplex resolutionREVIEWER COMMENTS

Reviewer #1 (Remarks to the Author):

A CK2 and SUMO-dependent, PML NB-involved regulatory mechanism controlling BLM ubiquitination and G quadruplex resolution

In this manuscript, Liu and colleagues uncover the mechanistic details of how BLM protein level is regulated by CK2-ARKL1-RNF111 axis in a PML-NB-dependent manner, and their connections to G4 resolution. Mainly, I think there are three new findings emerging from this study: 1) Based on previous finding that CK2 interacts with ARKL1, the authors further find CK2 phosphorylates ARKL1 at certain sites and facilitates the recruitment of SUMOylated RNF111 by ARKL1 SIM; 2) The authors find RNF111 is a SUMOylated protein, which has not been reported before. 3) Although BLM has been found to be ubiquitylated by MIB1 previously, the authors here find a second E3 ligase for BLM. In my opinion, the authors did a detailed and nice mechanistic study, although the physiological relevance of this mechanism to G4 or other genome stability-related structures can be further strengthened. The data in this manuscript are largely convincing and solid enough to support the authors' conclusions. But before granting publication, these are some issues the authors may need to address.

Major points:

1. The frequently considered functions of BLM in DNA damage repair field are facilitating DNA resection in HR and suppressing SCE instead of resolving G4. In the last part of this study, the authors connect the mechanism to G4 resolution. Could the authors explain why they specifically choose this angle to further explore the physiological relevance? Besides G4 resolution, performing experiments to evaluate the effects of CK2-ARKL1-RNF111-mediated BLM regulation on HR, SCE, genome stability, etc., would be more appealing.
2. As before this study, there already has been a publication in Mol Cell reporting MIB1 functions as an E3 ubiquitin ligase to BLM. Thus, I wonder the difference and connection between MIB1- and RNF111-mediated BLM ubiquitination. Using MIB1 as a control side by side to check whether MIB1 behaves like RNF111 to ubiquitylate BLM in PML NB or else, and discussing these in this paper may help the field to fully understand the regulatory mechanism of BLM ubiquitination.
3. Have the authors tried to investigate in which setting the CK2-ARKL1-RNF111-mediated BLM ubiquitination is controlled on or off? Could the G4 levels in cells or cell cycle initiate the fluctuation of CK2-axis-mediated BLM ub? Like that, in the published MIB1-BLM study, MIB1-mediated BLM ubiquitination occurs in G1 to facilitate NHEJ, while subsequently in S/G2, MIB1-BLM ub is suppressed to promote HR repair and SCE processing.

Minor points:

1. So far, factors controlling both RNF111 and BLM localization in PML NB are clear. But those affecting ARKL1 localization in PML NB are unclear, have the authors checked whether ARKL1 SIM motif, its interaction with or phosphorylation by CK2 control ARKL1 recruitment to PML NB?
2. It would be better to further figure out which K is the major SUMO site as the authors only use double K (K237R and K238R), as in Figure 3h. And is there any specific reason to check SUMO levels only under MG132 treatment?
3. Does RNF111 act like MIB1 to be capable of direct interacting with and ubiquitylating BLM in vitro? And it would be better to confirm RNF111 and its 2KR form SUMOylation in vitro.

4. I find in some figures RNF111 shows two bands and in others only one band? Is this due to that different cell lines are used or else unmentioned circumstances? If not, to be consistent, could the authors always show the two bands and have them indicated properly?

5. Figures with low quality or inconsistency: Figure 2c, ARKL1 panel. Figure 2c, BLM panel of siCtrl group vs that in Figure 1c; Figure 2e, RNF111 panel; and these two bands should both be RNF111, could the authors explain why only the above band is degraded?

Reviewer #2 (Remarks to the Author):

In this manuscript, Wang and colleagues demonstrate that BLM proteins are regulated by SUMO targeted ubiquitin ligase, RNF111. Furthermore, the G-quadruplex antibody signal reduced upon BLM upregulation when ARKL1, RNF111 were down-regulated or CK2 kinase was inhibited. The manuscript is well-written and includes a large number of experiments. There are several concerns that should be addressed prior to publication in Nature Communications.

The author mainly relies on measuring G4 signal intensity using IF, which is not entirely convincing. An alternative method would be to use Cut&Tag using BG4 antibody, which is more convincing. Lyu et al., 2021 Teng et al., 2021

BLM K317R, K331R may retain PML NB localization ability, since SIM-motif might be sufficient for BLM recruitment to PML NB. It should be checked whether BLM K317R/K331R can be recruited into PML NB.

According to the authors' data and model, it seems that RNF111 and BLM co-localized well in PML. It would be more convincing if there were data showing that the G4 that BLM unwinds is also located in the PML body.

In many WB, the Molecular weight (MW) is not indicated. The author should include the MW in all WB.

While the sumo-interacting motif (SIM) of RNF111 is not required for RNF111-BLM interaction, RNF111 SIM mutants can localize at PML bodies?

The RNF111-mediated ubiquitination of BLM occurs at PML bodies. How does PML KO/KD affect the RNF111 BLM protein status if RNF111-mediated ubiquitination occurs at PML bodies?

Fig.1c, Fig.1g, Fig. 1h, Fig. 2c, Fig. 2e, Fig. S3b, Fig. S3g: How many times were the experiments repeated? In reporting summary, that authors indicated that the experiments were all performed at least 2 times, but the quantification in those are based on single experiment?

It is unclear which PML isoform is used in the MSCV-GFP-PML plasmid.

Material methods: Where was siRNF111 #2 used?

Fig 1c,1g,1h: Are the signals normalized by GAPDH?

Fig4a, 4g, 5a, 6e, 6f: Adding scale bar is required.

Fig7: Since, they showed and counted G4 signal in only nucleus stained with DAPI, "cellular G4" might not be a proper word.

Fig7: As authors mentioned, the G4 could have regulatory roles. However, even considering that it is a U2OS cell, the G4 signal seems to be too high in WT at the representative pictures.

In addition, it would be desirable to show the level of DNA damage, such as the rH2AX signal. If the G4 level is under control, as the authors mention in the discussion, the DNA damage signal should not deviate much from the basal line.

Reviewer #3 (Remarks to the Author):

Lui and colleagues have submitted a study for review that characterizes the role of RNF111/Arkadia and related ARKL1 (also known as C18orf25) in the casein kinase 2 (CK2)-dependent regulation of SUMO-directed ubiquitination of the Bloom Syndrome protein BLM at PML nuclear bodies. They define the role of the CK2 phosphorylation of the SUMO interaction motif (SIM) of ARKL1 in recruitment of RNF111 to PML bodies, which in turn promotes the ubiquitination of BLM. Depletion or knockout of ARKL1 or RNF111 was consequently shown to lead to increased levels of BLM and a decrease in G-quadruplex DNA levels in cells (a major contributor to telomere structure). Several cell lines (U2OS, HeLa and BJ fibroblasts) and addback approaches using both wildtype and catalytically inactive RNF111 were employed, greatly increasing confidence in the role of this protein in the regulation of BLM ubiquitination and stability. Although the conclusions are supported by the data and the work is well done, there are a few instances where quantification and statistical analyses of Western blot data are missing for key figures (see major comments below). In addition, the study conclusions and results should be discussed in the context of Alternative Lengthening of Telomeres (ALT) as the U2OS cells they use are ALT and BLM plays a major role in the cell biology and mechanism of ALT.

Major Comments

- 1) BLM levels in RNF111 and ARKL1 knockdown and knockout cells was presented in figures 1 and 2 as qualitative data, there are no statistics. This data should be quantified and statistics provided on at least 3 replicates.
- 2) In Figure 5, for completeness, it would be good to show the localization of the SIM domain mutants of RNF111 relative to PML bodies, and as a control for BLM-RNF111sim PLA/PML co-localization.
- 3) Since the authors model involves ubiquitination of BLM at PML bodies, an orthogonal approach to increase confidence in this model would be to localize K48 ubiquitin with BLM (using PLA) at PML bodies in WT and RNF111/ARKL1 siRNA or KO cells, with WT and SIM domain mutant addbacks of RNF111/ARKL1.
- 4) For Figure 6, siRNA knockdown of CK2beta should be added to complement the CK2 inhibitor data as a second orthogonal approach.
- 5) Since BLM is important in regulating Alternative lengthening of telomeres (ALT) and that U2OS cells employ ALT to maintain their telomeres, the authors should look at both telomere protein localization to PML bodies (e.g. TRF1/2) to assess ALT-body formation, as well as assessing by PLA the colocalization of G-quadruplex and TRF1/2 (for telomeres) to demonstrate if there are specific disruption of ALT-bodies or G-quadruplexes at telomeres (respectively) when BLM stability is altered in RNF111/ARKL1 KD/KO cells.

Minor Comments

1) For better context for the reader in evaluating recovery times, the RFN111 and ARKL1 FRAP data should be discussed in the context of recovery times of PML and other PML body components (like SP100), see Brand et al., 2010 PMC Biophysics PMID: 20205709; and Weidtkamp-Peters et al., 2008 JCS PMID: 18664490

2) The Discussion should contain more information about the role of BLM, and G-quadruplexes in telomere maintenance and in ALT. Overall, the manuscript would benefit from better integration of ALT mechanisms given the main work horse cell line in the study, U2OS, maintain their telomeres by ALT.

Response to REVIEWER COMMENTS

Reviewer #1 (Remarks to the Author):

A CK2 and SUMO-dependent, PML NB-involved regulatory mechanism controlling BLM ubiquitination and G quadruplex resolution

In this manuscript, Liu and colleagues uncover the mechanistic details of how BLM protein level is regulated by CK2-ARKL1-RNF111 axis in a PML-NB-dependent manner, and their connections to G4 resolution. Mainly, I think there are three new findings emerging from this study: 1) Based on previous finding that CK2 interacts with ARKL1, the authors further find CK2 phosphorylates ARKL1 at certain sites and facilitates the recruitment of SUMOylated RNF111 by ARKL1 SIM; 2) The authors find RNF111 is a SUMOylated protein, which has not been reported before. 3) Although BLM has been found to be ubiquitylated by MIB1 previously, the authors here find a second E3 ligase for BLM. In my opinion, the authors did a detailed and nice mechanistic study, although the physiological relevance of this mechanism to G4 or other genome stability-related structures can be further strengthened. The data in this manuscript are largely convincing and solid enough to support the authors' conclusions. But before granting publication, these are some issues the authors may need to address.

Major points:

1. The frequently considered functions of BLM in DNA damage repair field are facilitating DNA resection in HR and suppressing SCE instead of resolving G4. In the last part of this study, the authors connect the mechanism to G4 resolution. Could the authors explain why they specifically choose this angle to further explore the physiological relevance? Besides G4 resolution, performing experiments to evaluate the effects of CK2-ARKL1-RNF111-mediated BLM regulation on HR, SCE, genome stability, etc., would be more appealing.

BLM helicase is highly active in binding and unwinding G4 structures (Sun et al., 1998; PMID:9765292; Huber et al., 2002; PMID:12235379); and G4s naturally occur through the genome playing critical roles in many aspects of cellular processes. Our study focuses on the physiological importance of the RNF111-ARKL1-dependent control of BLM in unperturbed cells. The assays for the role of BLM in HR, SCE and genome stability often involves DNA damage. The regulation of BLM in response to DNA damage likely involves additional aspects of regulation.

2. As before this study, these already has been a publication in Mol Cell reporting MIB1 functions as an E3 ubiquitin ligase to BLM. Thus, I wonder the difference and connection between MIB1- and RNF111-mediated BLM ubiquitination. Using MIB1 as a control side by side to check whether MIB1 behaves like RNF111 to ubiquitylate BLM in PML NB or else, and discussing these in this paper may help the field to fully understand the regulatory mechanism of BLM ubiquitination.

In the above-mentioned publication (Wang et al, 2013, PMID:24239288), it showed that TOPBP1 depletion led to reduced BLM levels and depletion of MIB1 restored BLM protein levels in TOPBP1-deficient cells. It also showed that MIB1 overexpression increased BLM ubiquitination and MIB1 ubiquitinates BLM *in vitro*. However, the effect of depletion of MIB1 alone on BLM levels was not addressed in the publication. We found that depletion of MIB1, unlike that of RNF111, does not lead to elevated BLM protein levels. It also does not affect BLM protein levels when combined with depletion of RNF111 (data shown here). Considering that a later publication has shown that depletion of TOPBP1 does not lead to degradation of BLM (Blackford, et al, 2015, PMID:25794620), it remains unclear whether and how MIB1 is involved in regulating BLM protein stability. The research on MIB1's function is not relevant to the scope of our manuscript; we therefore prefer not to include this data for discussion in our manuscript.

Fig. Depletion of MIB1 does not affect BLM protein levels. U2OS cells were transfected with indicated siRNAs

3. Have the authors tried to investigate in which setting the CK2-ARKL1-RNF111-mediated BLM ubiquitination is controlled on or off? Could the G4 levels in cells or cell cycle initiate the fluctuation of CK2-axis-mediated BLM ub? Like that, in the published MIB1-BLM study, MIB1-mediated BLM ubiquitination occurs in G1 to facilitate NHEJ, while subsequently in S/G2, MIB1-BLM ub is suppressed to promote HR repair and SCE processing.

Our study has identified a novel regulatory mechanism controlling BLM protein levels and demonstrated its effect on G quadruplex regulation in unperturbed cells. BLM is a highly regulated protein which undergoes multiple forms of posttranslational modifications in response to DNA damage or during cell cycle. The additional regulation of the pathway and its interaction with additional regulations of BLM may be investigated in the future.

Minor points:

1. So far, factors controlling both RNF111 and BLM localization in PML NB are clear. But those affecting ARKL1 localization in PML NB are unclear, have the authors checked whether ARKL1 SIM motif, its interaction with or phosphorylation by CK2 control ARKL1 recruitment to PML NB?

It has been shown that ARKL1^{ΔS} mutant which lacks the interaction with CK2 can still be recruited to PML NBs (Cao et al., 2014; PMID:24216761).

Many PML NBs associated proteins required their SIM domains to localize to PML NBs (e.g. BLM, RNF111). We found that ARKL1 also requires its SIMs to localize to PML NBs and ARKL1^{SIM*} mutant fails to localize to PML NBs. This is likely due to the reason that the SIM mutation disrupts the binding of ARKL1 with PML. These data are now included as Supplemental Fig. 4e, 4f. The SIM-dependent recruitment of ARKL1 to PML NBs is independent of RNF111 since ARKL1's localization to PML NBs is not affected in RNF111 KO cells (Supplemental Fig. 4d). The recruitment of ARKL1 to PML NBs may be further investigated in the future.

2. It would be better to further figure out which K is the major SUMO site as the authors only use double K (K237R and K238R), as in Figure 3h. And is there any specific reason to check SUMO levels only under MG132 treatment?

Mutation of both the residues (K237R/K238R) but not each individual residue (K237R or K238R) reduced RNF111 SUMOylation, indicating that both residues are required for proper SUMOylation. The additional data for the effect of K237R and K238R are now included as Supplemental Fig.3h and 3i.

MG132 has no effect on the SUMOylation. An experiment performed without MG132 gives similar results (shown here, please compare it with **Fig. 3h**). In order to avoid the confusion, we removed the MG132 label in Main **Fig.3f and 3h**. Instead, we described it in the methods that SUMOylation experiment with cells treated or not treated with MG132 gives similar results.

Fig. RNF111^{K218/237R} mutant exhibits reduced SUMOylation. The IPs were performed under denature condition using lysates of 293T cells expressing indicated constructs.

3. Does RNF111 act like MIB1 to be capable of direct interacting with and ubiquitylating BLM *in vitro*? And it would be better to confirm RNF111 and its 2KR SUMOylation *in vitro*.

We have performed an *in vitro* ubiquitination experiment to examine whether RNF111 can ubiquitinate purified BLM *in vitro*. As shown in Supplemental Fig. 1c, purified immunoprecipitated RNF111-Flag WT but not CS mutant ubiquitinates purified GFP-BLM. The SUMOylation of RNF111 and the effect of K237/238R mutation on the SUMOylation are evident in the experimental data that we have shown (Fig. 3 and Supplemental Fig.3).

4. I find in some figures RNF111 shows two bands and in others only one band? Is this due to that different cell lines are used or else unmentioned circumstances? If not, to be consistent, could the authors always show the two bands and have them indicated properly?

In U2OS cells, RNF111 appears to have two bands with the top band around the size of 150 KD (as shown in **Fig. 1a**). The lower band is often weak to be detected in other cell lines (e.g. HeLa, BJ and 293T cells).

5. Figures with low quality or inconsistency: Figure 2c, ARKL1 panel. Figure 2c, BLM panel of siCtrl group vs that in Figure 1c; Figure 2e, RNF111 panel; and these two bands should both be RNF111, could the authors explain why only the above band is degraded?

Fig. 2c, ARKL1 panel is replaced with a longer exposure for the band in siRNA-treated samples to be seen.

Fig. 2c, BLM level is now quantified with data from three independent experiments and shown with statistics.

Fig. 2e, the RNF111 lower band (or isoform) is likely to be more stable and did not show degradation at 1h in the experiment. We quantified the RNF111 full-length (top band) as indicated by the arrow in the panel.

Reviewer #2 (Remarks to the Author):

In this manuscript, Wang and colleagues demonstrate that BLM proteins are regulated by SUMO targeted ubiquitin ligase, RNF111. Furthermore, the G-quadruplex antibody signal reduced upon BLM upregulation when ARKL1, RNF111 were down-regulated or CK2 kinase was inhibited. The manuscript is well-written and includes a large number of experiments. There are several concerns that should be addressed prior to publication in Nature Communications.

The author mainly relies on measuring G4 signal intensity using IF, which is not entirely convincing. An alternative method would be to use Cut&Tag using BG4 antibody, which is more convincing. Lyu et al., 2021 Teng et al., 2021

In our study, we have used the BG4 antibody (Biffi et al., 2013; PMID:23422559) for the staining of G4s in the cells. The BG4 antibody staining specificity was supported by our results in Supplemental Fig. 7a showing that BG4 antibody staining can readily detect the increased G4 levels induced by Pyridostatin (PDS), an established G4 stabilizer. The BG4 antibody has been widely used for detecting G4s in cells (e.g. Drosopoulos et al., 2015; PMID:26195664); Yadav, et al., 2022; PMID:36265486). We appreciate the suggestion from the reviewer to utilize G4 CUT &Tag to analyze genome-wide G4s. However, the studies mentioned by the reviewer which perform G4 CUT&Tag (Lyu et al., 2021; Teng et al., 2021) also used BG4 antibody. To be more confident on the effect of G4 levels upon loss of RNF111 and ARKL1-dependent control of BLM, we used another established G4-specific antibody, 1H6 (Henderson, 2014; PMID:24163102). In Supplemental Fig. 7h, 7i, G4 staining with 1H6 antibody also showed that depletion of ARKL1 or RNF111 resulted in reduced G4 levels, like the results obtained with G4 staining with BG4 antibody.

BLM K317R, K331R may retain PML NB localization ability, since SIM-motif might be sufficient for BLM recruitment to PML NB. It should be checked whether BLM K317R/K331R can be recruited into PML NB.

It has been reported that BLM K317R/K331R is compromised in its ability to localize to PML NBs (Eladad et al., 2005; PMID:15829507). We confirmed this and the data is now included in Supplemental Fig. 5e.

According to the authors' data and model, it seems that RNF111 and BLM co-localized well in PML. It would be more convincing if there were data showing that the G4 that BLM unwinds is also located in the PML body.

In our model, RNF111-dependent BLM ubiquitination involves PML NBs, however, the effect of this regulation on G4 is likely to be genome wide. This is supported by our data showing that

G4 levels in the nucleus is largely reduced in RNF111- or ARKL1-deficient cells (Fig.7 and Supplemental Fig.7).

In many WB, the Molecular weight (MW) is not indicated. The author should include the MW in all WB.

The molecular weight is now included in the figures.

While the sumo-interacting motif (SIM) of RNF111 is not required for RNF111-BLM interaction, RNF111 SIM mutants can localize at PML bodies?

It has been reported that the SIM domains are required for RNF111's localization to PML NBs and RNF111 SIM* mutant is deficient in localizing to PML NBs (Erker et al., 2013; PMID:23530056). We confirmed this and the data is now included in Supplemental Fig. 5d.

The RNF111-mediated ubiquitination of BLM occurs at PML bodies. How does PML KO/KD affect the RNF111 BLM protein status if RNF111-mediated ubiquitination occurs at PML bodies?

Depletion of PML also leads to increased BLM protein levels. This data is now included as Supplemental Fig. 5c.

Fig.1c, Fig.1g, Fig. 1h, Fig. 2c, Fig. 2e, Fig. S3b, Fig. S3g: How many times were the experiments repeated? In reporting summary, that authors indicated that the experiments were all performed at least 2 times, but the quantification in those are based on single experiment?

For Fig. 1c, 2c, 2e, the revised figure now represents quantification from three independent experiments.

For Fig. 1g, 1h, S3b, S3g, the quantification is from two independent experiments.

It is unclear which PML isoform is used in the MSCV-GFP-PML plasmid.

PML isoform II is used in the plasmid. This information is now included in the Methods.

Material methods: Where was siRNF111 #2 used?

We thank the reviewer for catching this mistake! Since we didn't show any data of siRNF111#2, we deleted this in the Methods section.

Fig 1c,1g,1h: Are the signals normalized by GAPDH?

For Fig. 1c, the relative BLM level at each time point normalized by GAPDH and relative to that at 0 h is quantified.

For Fig. 1g, the relative BLM level in the HA immunoprecipitates (normalized by immunoprecipitated HA-RNF111) is quantified.

For Fig. 1h, the relative RNF111 level present in the GFP immunoprecipitates (normalized by immunoprecipitated GFP-BLM) is quantified.

These descriptions are now included in the revised Figure Legends.

Fig4a, 4g, 5a, 6e, 6f: Adding scale bar is required.

Scale bars are now included in these figures.

Fig7: Since, they showed and counted G4 signal in only nucleus stained with DAPI, “cellular G4” might not be a proper word.

This is now corrected.

Fig7: As authors mentioned, the G4 could have regulatory roles. However, even considering that it is a U2OS cell, the G4 signal seems to be too high in WT at the representative pictures.

We have used two different G4 specific antibodies and similar results were obtained. The additional G4 1H6 antibody staining is now included in Supplemental Fig. 7h, i. The experiments were also performed with BJ cells and the effect of depletion of RNF111 and ARKL1 are similar (Supplemental Fig. 7g, i).

In addition, it would be desirable to show the level of DNA damage, such as the γ H2AX signal. If the G4 level is under control, as the authors mention in the discussion, the DNA damage signal should not deviate much from the basal line.

Depletion of RNF111 or ARKL1 did not result in elevated γ H2AX levels, indicating that loss of RNF111- and ARKL1-dependent control of BLM does not induce excessive DNA damage. This data is now shown in Supplemental Fig. 7f.

Reviewer #3 (Remarks to the Author):

Lui and colleagues have submitted a study for review that characterizes the role of RNF111/Arkadia and related ARKL1 (also known as C18orf25) in the casein kinase 2 (CK2)-dependent regulation of SUMO-directed ubiquitination of the Bloom Syndrome protein BLM at PML nuclear bodies. They define the role of the CK2 phosphorylation of the SUMO interaction motif (SIM) of ARKL1 in recruitment of RNF111 to PML bodies, which in turn promotes the ubiquitination of BLM. Depletion or knockout of ARKL1 or RNF111 was consequently shown to lead to increased levels of BLM and a decrease in G-quadruplex DNA levels in cells (a major contributor to telomere structure). Several cell lines (U2OS, HeLa and BJ fibroblasts) and addback approaches using both wildtype and catalytically inactive RNF111 were employed, greatly increasing confidence in the role of this protein in the regulation of BLM ubiquitination and stability. Although the conclusions are supported by the data and the work is well done, there are a few instances where quantification and statistical analyses of Western blot data are missing for key figures (see major comments below). In addition, the study conclusions and results

should be discussed in the context of Alternative Lengthening of Telomeres (ALT) as the U2OS cells they use are ALT and BLM plays a major role in the cell biology and mechanism of ALT.

We have addressed the issues raised about quantification and statistical analyses. We also have included a discussion of ALT in the discussion.

Major Comments

1) BLM levels in RNF111 and ARKL1 knockdown and knockout cells was presented in figures 1 and 2 as qualitative data, there are no statistics. This data should be quantified and statistics provided on at least 3 replicates.

The data are now quantified with biological replicates and with statistics. See revised Fig. 1c, 2c, 2e and corresponding figure legends.

2) In Figure 5, for completeness, it would be good to show the localization of the SIM domain mutants of RNF111 relative to PML bodies, and as a control for BLM-RNF111sim PLA/PML co-localization.

It has been shown previously that the RNF111 SIM* mutant fails to localize to PML NBs (Erker et al., 2013; PMID:23530056). We confirmed this and the data is now included as Supplemental 5d.

3) Since the authors model involves ubiquitination of BLM at PML bodies, an orthogonal approach to increase confidence in this model would be to localize K48 ubiquitin with BLM (using PLA) at PML bodies in WT and RNF111/ARKL1 siRNA or KO cells, with WT and SIM domain mutant addbacks of RNF111/ARKL1.

We appreciate the reviewer's suggestion. However, K48 ubiquitin-modified BLM may undergo rapid degradation for a detection by PLA. In addition, the K48 antibody does not work well in our experience.

4) For Figure 6, siRNA knockdown of CK2beta should be added to complement the CK2 inhibitor data as a second orthogonal approach.

Like the effect of CK2 inhibitor, knockdown of CK2beta abolished the phosphorylation of ARKL1 by CK2 (Supplemental Fig. 6e). It also leads to reduced BLM ubiquitination and elevated BLM protein levels (Supplemental Fig. 6f, g).

5) Since BLM is important in regulating Alternative lengthening of telomeres (ALT) and that U2OS cells employ ALT to maintain their telomeres, the authors should look at both telomere protein localization to PML bodies (e.g. TRF1/2) to assess ALT-body formation, as well as assessing by PLA the colocalization of G-quadruplex and TRF1/2 (for telomeres) to demonstrate if there are specific disruption of ALT-bodies or G-quadruplexes at telomeres (respectively) when BLM stability is altered in RNF111/ARKL1 KD/KO cells.

Our data have indicated that the RNF111- and ARKL1-dependent control of BLM protein levels occurs in both ALT cells (e.g. U2OS) and non-ALT cells (e.g. HeLa, BJ). The effect of disruption of this control on G-quadruplex in the nucleus is similar in both ALT U2OS cells and non-ALT BJ cells. We appreciate the comments of the reviewer on investigating the importance of this regulatory control of BLM in ALT. However, this is beyond the scope of this manuscript and will be investigated in the future.

Minor Comments

1) For better context for the reader in evaluating recovery times, the RNF111 and ARKL1 FRAP data should be discussed in the context of recovery times of PML and other PML body components (like SP100), see Brand et al., 2010 PMC Biophysics PMID: 20205709; and Weidtkamp-Peters et al., 2008 JCS PMID: 18664490

We thank the reviewer for the references! However, we are hesitant to compare our FRAP results on RNF111 and ARKL1 with those in the references on PML isoforms and some components of the PML NBs, such as SP100. The differences in experimental condition, duration of the live-cell imaging, cell lines, etc., are likely to influence on the results and comparison. For example, the PML isoform that shows the fastest recovery time is different in the two studies mentioned above. In addition, the measurement and calculation in the references were performed differently without having a half-time recovery value for their samples that we can use for a comparison with our FRAP data. We believe that a comparison will only be meaningful when samples are examined in the same experimental setting.

2) The Discussion should contain more information about the role of BLM, and G-quadruplexes in telomere maintenance and in ALT. Overall, the manuscript would benefit from better integration of ALT mechanisms given the main work horse cell line in the study, U2OS, maintain their telomeres by ALT.

We appreciate the reviewer's suggestion! Our data indicate that the RNF111-ARKL1-dependent control of BLM and its effect on G4 levels occur in both ALT (e.g. U2OS) and non-ALT (e.g. BJ) cells. We have included a discussion of a possible role of RNF111-ARKL1-dependent control of BLM in telomere maintenance and ALT in the text.

REVIEWERS' COMMENTS

Reviewer #1 (Remarks to the Author):

The authors have addressed the concerns I raised. I am satisfied with this revision.

PS: In the rebuttal letter, "As shown in Supplemental Fig. 1c, purified immunoprecipitated RNF111-Flag WT but not CS mutant ubiquitinates purified GFP-BLM." Supplemental Fig. 1c should be 1b.

Reviewer #2 (Remarks to the Author):

The author has made notable strides in addressing the reviewers' concerns, presenting a comprehensive set of additional data. Yet, the incorporation of the 1H6 antibody as a substitute for the immunostaining-based technique fell short of the expected standards. A significant highlight of this manuscript is the quantification of G4 in relation to BLM/RNF111-dependency. This was a focal point of concern for this reviewer, who had initially advocated for an alternative quantification method, specifically cut-and-tag. Additionally, the distinct substrate profiles of the 1H6 and BG4 antibodies warrant attention and remain an aspect not thoroughly addressed. Nonetheless, with the reservations in mind, this reviewer believes the overall quality and depth of the revisions suggest that the manuscript is suitable for publication in Nature Communications. But the decision will be made by overall comments by other reviewers and editor's decision.

Reviewer #3 (Remarks to the Author):

The reviewers concerns were addressed.

Responses to REVIEWERS' COMMENTS

We thank the reviewers for their critical points for us to improve the manuscript!

Reviewer #1 (Remarks to the Author):

The authors have addressed the concerns I raised. I am satisfied with this revision.

PS: In the rebuttal letter, "As shown in Supplemental Fig. 1c, purified immunoprecipitated RNF111-Flag WT but not CS mutant ubiquitinates purified GFP-BLM." Supplemental Fig. 1c should be 1b.

We thank the reviewer for the point!

Reviewer #2 (Remarks to the Author):

The author has made notable strides in addressing the reviewers' concerns, presenting a comprehensive set of additional data. Yet, the incorporation of the 1H6 antibody as a substitute for the immunostaining-based technique fell short of the expected standards. A significant highlight of this manuscript is the quantification of G4 in relation to BLM/RNF111-dependency. This was a focal point of concern for this reviewer, who had initially advocated for an alternative quantification method, specifically cut-and-tag. Additionally, the distinct substrate profiles of the 1H6 and BG4 antibodies warrant attention and remain an aspect not thoroughly addressed. Nonetheless, with the reservations in mind, this reviewer believes the overall quality and depth of the revisions suggest that the manuscript is suitable for publication in Nature Communications. But the decision will be made by overall comments by other reviewers and editor's decision.

We thank the reviewer for the point and appreciate the suggestion! Both BG4 and 1H6 antibodies have been widely used in immunofluorescence to detect G4 abundance. Although it is not clear whether these antibodies recognize distinct types of G4s, it has been shown that both these antibodies detect genuine G4s in immunofluorescence or ChIP. For example, several studies have used both antibodies to verify G4 signals detected by each other in immunofluorescence or ChIP (PMID: 30808951; PMID:33758195; PMID:33953191). In our study, G4 staining results by BG4 is confirmed by 1H6 staining results, further supporting the effect of RNF111-ARKL1-dependent regulation of BLM on G4 levels.

Reviewer #3 (Remarks to the Author):

The reviewers concerns were addressed.